# Inference-Time Alignment via Hypothesis Reweighting

**Yoonho Lee, Jonathan Williams, Henrik Marklund, Archit Sharma,**
**Eric Mitchell, Anikait Singh, Chelsea Finn**
*Stanford University*
*Correspondence: yoonho@cs.stanford.edu*

**Reviewed on OpenReview:** *https: // openreview. net/ forum? id= Q9p8LSEpiJ*

## Abstract

Reward models trained on aggregate preferences often fail to capture individual users' values, but existing adaptation methods such as fine-tuning or long-context conditioning are too costly for real-time personalization. We propose Hypothesis Reweighting (HyRe), which enables real-time personalization by reweighting ensemble members using just 1-5 labeled examples from the target user or domain. Our method builds on the empirical observation that when different heads capture different valid interpretations of preference data, reweighting them can substantially outperform uniform averaging. HyRe trains a single network with multiple prediction heads that capture different valid interpretations of preference data, then uses a Bayesian update to upweight the heads that best match the target user's preferences. This requires only a single forward pass with negligible ($<1\%$) computational overhead, making it practical for inference-time personalization. We evaluate HyRe across diverse target preference distributions. With as few as five preference pairs per target distribution, HyRe surpasses state-of-the-art reward models on RewardBench at 2B and 8B scale and improves reward model accuracy by 20% across 32 personalization tasks.

## 1 Introduction

Task specification—describing precisely what a machine learning model should do—is inherently iterative and fundamentally incomplete under any finite set of instructions or training examples. As models grow more powerful and are applied to increasingly complex and nuanced tasks, this problem is pronounced for preference learning, where different users have conflicting notions of desirable behavior. Consider a chatbot trained via Reinforcement Learning from Human Feedback (RLHF) [62] on a broad distribution of user preferences. Such models often perform adequately in aggregate but systematically fail to address specific users' needs, since different individuals have distinct, sometimes contradictory, notions of desirable responses. Meeting these user-specific requirements necessitates rapid model adaptation with minimal supervision.

However, existing adaptation strategies face practical constraints: fine-tuning requires hundreds of gradient steps and risks catastrophic forgetting [28, 29, 48, 74], while in-context learning methods scale poorly with context length and lack sample efficiency in few-shot regimes [22, 37, 77]. For applications requiring sub-second per-user adaptation, such as large-scale personalized assistants, these constraints become critical bottlenecks. This motivates inference-time methods that can adapt without gradient updates or long contexts.

To efficiently resolve ambiguity at test time, we draw on recent progress in efficient ensemble architectures [50]. These methods let a single backbone network represent a broad range of plausible functions at low overhead, corresponding to different ways the model can interpret the training set. While prior work focuses largely on using ensembles for uncertainty estimation, we instead use them to disambiguate tasks in real time: by quickly identifying which members best match a new distribution, we can recover the interpretation most appropriate for that setting.

We introduce Hypothesis Reweighting (HYRE), a two-step approach that scales to large models. First, we train an ensemble of function heads on top of a shared backbone, ensuring each head individually fits the

Figure 1: **Overview of HyRe.** We train multiple prediction heads on a shared backbone. (Left) At inference time, we score each head on a small labeled adaptation set drawn from the target distribution. (Right) We apply a generalized Bayesian update (1) to upweight the heads that best match the adaptation set, then use the resulting weighted ensemble to make predictions on new inputs.

training data. Next, at inference time, we gather a few labeled examples from the target distribution—either proactively queried or provided in advance—and measure each head's performance. We then reweight the ensemble using a generalized Bayesian update that favors the heads performing best on the adaptation set. Crucially, this update supports non-differentiable metrics like 0-1 error and requires only a single forward pass over the adaptation set, making it far more efficient than conventional fine-tuning.

Our key contributions are as follows. **(1)** We introduce HYRE, a simple and efficient method for inference-time adaptation that reweights ensemble members based on a few labeled examples from the target distribution. **(2)** We show that uniform ensemble averaging can fail when different heads capture different valid interpretations of the data, motivating our adaptive reweighting approach. **(3)** We achieve state-of-the-art results on RewardBench at both 2B and 8B scales with just 1-5 adaptation examples, outperforming much larger models. **(4)** We show that the method scales to large models with negligible overhead ($< 1\%$) and improves performance across 32 preference-learning tasks.

## 2 Preliminaries

**Problem setup.** We consider a general supervised learning setting that includes classification, preference learning, and regression tasks. Let $\mathcal{X}$ represent the input space and $\mathcal{Y}$ the output space, with training distribution $P_{\text{train}}$ and evaluation distribution $P_{\text{eval}}$ defined over $\mathcal{X} \times \mathcal{Y}$. The training dataset $\mathcal{D}_{\text{train}} = \{(x_i, y_i)\}_{i=1}^N$ consists of $N$ examples drawn from $P_{\text{train}}$. We explore few-shot adaptation settings such as chatbot personalization, where a small adaptation set $\mathcal{D}_{\text{adapt}} \sim P_{\text{eval}}$ only partially informs model performance under $P_{\text{eval}}$. The adaptation set $\mathcal{D}_{\text{adapt}}$ can be labeled in advance or actively queried, and is much smaller than the training set ($|\mathcal{D}_{\text{adapt}}| \ll |\mathcal{D}_{\text{train}}|$). For instance, in our main experiment, $|\mathcal{D}_{\text{adapt}}| = 16$ compared to $|\mathcal{D}_{\text{train}}| > 300,000$, with adaptation occurring near-instantly after a single forward pass through the network.

**Ensemble architectures.** We train an ensemble of $K$ models $f_1, \ldots, f_K$ on the training data $\mathcal{D}_{\text{train}}$. We consider parameterizations of the ensemble that aim to represent a distribution over functions by training multiple models on the same dataset $\mathcal{D}_{\text{train}}$, ensuring diversity without computational overhead beyond training a single model. To achieve this, we employ *prior networks* [50]: fixed, randomly initialized models whose outputs are added to each ensemble member's output. This mechanism preserves diversity among ensemble members during training, even as individual models converge. We consider two computationally efficient ensemble architectures:

1. **Shared-Base Ensemble**: A single neural network that parameterizes both the prior and ensemble components by sharing a common base.

2. **Epinet**: A base network augmented by a small auxiliary network that introduces diversity via a learned index.

We train all ensemble members jointly by minimizing $\sum_{k=1}^K \mathcal{L}(f_k, \mathcal{D}_{\text{train}})$ using SGD. Note that this simple training procedure does not explicitly encourage diversity and may lead to ensemble collapse in some settings. However, we empirically observe that sufficient diversity emerges for personalization tasks, likely due to random initialization and the high-dimensional parameter space of large models. These architectures have negligible overhead—in our reward model experiments, 100 ensemble heads add only 550K parameters (0.03%) to the 2B-parameter Gemma backbone. Please see Section E for architectural details.

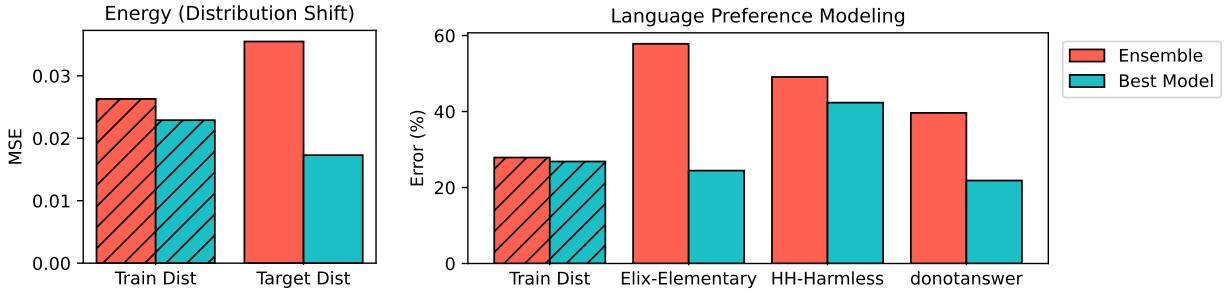

Figure 2: Motivating observation: the ensemble average often performs worse than a single well-chosen member. This tendency is particularly pronounced further away from the training distribution. HYRE **goes beyond selecting a single head by learning a continuous weighting over all heads**.

## 3 Inference-Time Ensemble Recalibration

In this section, we describe Hypothesis Reweighting (HYRE), a simple and computationally efficient method for few-shot adaptation that reweights ensemble members at test time using a small labeled set from the target task, without retraining any model parameters.

### 3.1 Uniform Ensemble Averaging is Often Suboptimal

A long-held principle in machine learning is that a uniformly weighted ensemble of independently trained models tends to outperform individual models, largely because averaging can smooth out errors and mitigate individual biases [15, 25, 39, 40, 52]. By exploiting diversity among ensemble members, the uniform average often reduces variance and improves predictive performance in settings where the training data sufficiently specifies the task at hand.

However, settings with substantial distribution shifts or personalized requirements can render the uniform ensemble average overly coarse. When ensemble members represent different valid interpretations of the training data, averaging dilutes these meaningful distinctions. Recent work [43, 69] shows that in such cases, selecting the best single model can outperform naive averaging, as individual models may better align with specific test conditions.

While prior studies have shown this phenomenon in controlled or synthetic tasks, we verify in Figure 2 that it persists in large-scale, real-world settings involving distribution shifts and personalization. Uniform ensemble predictions indeed fall short of even the accuracy achieved by a suitably chosen single model.

This observation motivates our approach: *reweight the ensemble according to how well each head aligns with the target task*. Using a small labeled set from the target distribution, we upweight the prediction heads that best match target data. We detail this reweighting procedure in Section 3.2 and validate its effectiveness on real-world tasks in Section 6.

> **Key insight: Uniform averaging can blur together distinct hypotheses**
>
> When different ensemble members capture different valid interpretations of the data, uniform averaging can wash out the one that best matches the target context. HYRE instead reweights the ensemble toward the interpretations that best align with a small labeled adaptation set.

### 3.2 Fast Inference-Time Ensemble Reweighting (HyRe)

Given an ensemble of $K$ models $f_1, \ldots, f_K$, we aim to dynamically update their weights based on adaptation data. As a practical inference-time assumption in settings where we cannot further train neural networks, we can think of the "best" model as being one of the $K$ ensemble particles that performs best on the evaluation distribution. Starting with uniform weights $w_k = \frac{1}{K}$, we update them as new labeled data from $\mathrm{P}_{\text{eval}}$ becomes available.

The weighted ensemble prediction is $f_w(x) = \sum_{k=1}^{K} w_k f_k(x)$, where each $w_k \geq 0$ and $\sum_{k=1}^{K} w_k = 1$. We measure each member's performance using a loss function $l(f_k, x, y)$ and compute their cumulative loss on adaptation data $\mathcal{L}(f_k, \mathcal{D}_{\text{adapt}}) = \sum_{(x,y) \in \mathcal{D}_{\text{adapt}}} l(f_k, x, y)$. The weights are updated using a softmax on

---

**Algorithm 1** HYRE (Inference Time)

---

**Require:** Ensemble members $f_1, \ldots, f_K$, unlabeled pool $\{x_n\}_{n=1}^M$, query budget $B$
  1: Initialize weights $w_k \leftarrow 1/K$ for all $k$, adaptation set $Q \leftarrow \emptyset$
  2: **for** $i \leftarrow 1$ to $B$ **do**
  3:     Select $x_n$ (randomly, or via $\arg\max_n c(x_n)$ where $c$ measures ensemble disagreement; Section C)
  4:     Query label $y_n$; update $Q \leftarrow Q \cup \{(x_n, y_n)\}$
  5:     Compute loss $\mathcal{L}_k \leftarrow \sum_{(x,y) \in Q} l(f_k(x), y)$ for each $k$                    (Section 3.2)
  6:     Update weights $w_k \propto \exp(-\mathcal{L}_k)$, normalized so $\sum_k w_k = 1$
  7: **end for**
  8: **Return** weighted ensemble $f_w : x \mapsto \sum_{k=1}^K w_k f_k(x)$

---

negative cumulative loss:

$$w_k = \frac{\exp(-\mathcal{L}(f_k, \mathcal{D}_{\text{adapt}}))}{\sum_{j=1}^K \exp(-\mathcal{L}(f_j, \mathcal{D}_{\text{adapt}}))}. \tag{1}$$

This update has a natural interpretation as generalized Bayesian inference (Section 3.3), where we treat each ensemble member as a hypothesis and update beliefs based on target-domain performance rather than likelihood. As the loss $l(f_k, x, y)$, we use 0-1 error for classification and mean squared error for regression, though HYRE supports any performance metric since the weight update remains valid for non-differentiable functions.

The complete adaptation procedure is summarized in Algorithm 1. Key practical considerations include:

- **Computational efficiency.** Since we use efficient ensemble architectures (Section 2), training and inference cost is comparable to that of a single network. Reweighting requires just one forward pass.

- **Coverage of $f_1, \ldots, f_K$.** The target function need not lie exactly in the ensemble's linear span. Restricting solutions to the ensemble's convex hull provides a practical bias-variance tradeoff in low-data regimes.

- **Active selection of $\mathcal{D}_{\text{adapt}}$.** When unlabeled samples are available at test time, we can actively select which to label, improving reweighting efficiency (Section C).

### 3.3 Interpreting HYRE as Generalized Bayesian Inference

The weight update in (1) can be interpreted as a form of generalized Bayesian inference [7]. Given an initial belief state $\pi(w)$, the updated belief after observing $\mathcal{D}_{\text{adapt}}$ is:

$$\pi(w | \mathcal{D}_{\text{adapt}}) \propto \exp\left(-\mathcal{L}(w, \mathcal{D}_{\text{adapt}})\right) \pi(w), \tag{2}$$

which generalizes classical Bayesian inference by allowing arbitrary loss functions. Standard Bayes is recovered when $l(w, x)$ is the negative log-likelihood.

Bissiri et al. [7] show that this update is the unique coherent way to update beliefs from a loss function, assuming only basic rationality axioms (e.g., if the loss is uninformative, the posterior equals the prior). This result applies to arbitrary loss functions, including non-differentiable metrics like 0-1 error, without requiring a correctly specified likelihood.

Standard results for generalized posteriors imply concentration on expected-loss minimizers as $|\mathcal{D}_{\text{adapt}}|$ grows under i.i.d. data and bounded loss. In our discrete setting with $K$ heads, this means that the weight concentrates on the best-performing heads, or spreads across equally good heads. Bounded loss is empirically important: with unbounded cross-entropy, a single mislabeled or outlier example can drive a head's weight near zero, and Table 5 confirms that cross-entropy degrades reweighting performance [31]. In some experiments, adaptation data is drawn i.i.d. from a target distribution that differs from the training distribution. In these cases, the i.i.d. assumption still holds, but concentration guarantees apply only relative to the best head in the ensemble. If the target lies outside the ensemble's span, reweighting cannot recover it. Table 2 shows consistent gains in these shifted settings, suggesting that the ensemble's coverage is sufficient in practice for the preference tasks we study. When the optional active data selection variant is used (Section C), adaptation

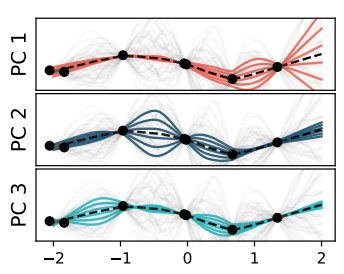

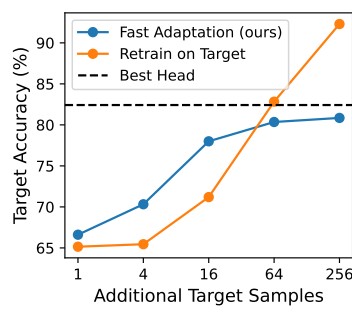

Figure 3: Principal component analysis of an ensemble of regression models. Left: Each gray line is the prediction of an ensemble member; the dashed line shows the ensemble mean. Right: The top three principal components of the ensemble's predictions reveal distinct axes of variation in predictive behavior. **Searching among ensemble weights like HYRE acts as a strong inductive bias towards simple functions consistent with the training data**.

Figure 4: HYRE vs. fine-tuning with different amounts of adaptation data. Despite using only a single forward pass, HYRE **outperforms fine-tuning in low-data regimes**.

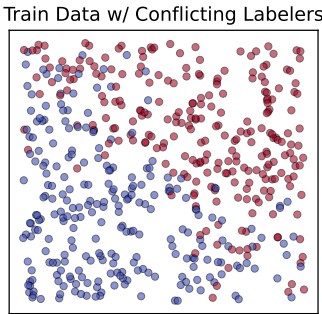

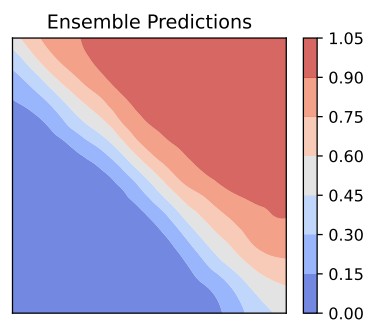

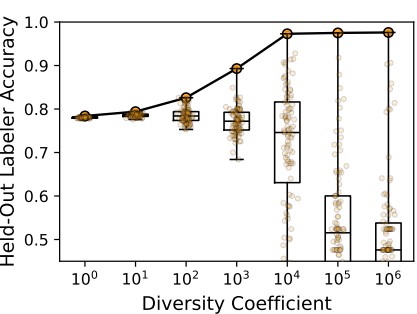

Figure 5: Ensemble behavior under label ambiguity. (Left) We simulate conflicting preferences between labelers on a synthetic dataset. (Center) Ensemble-averaged predictions approximate the consensus, smoothing over disagreements. (Right) We measure the maximum agreement between an ensemble and a held-out labeler: **increasing model diversity improves alignment with individual labelers**.

points are no longer strictly i.i.d. In practice, however, this only changes which points are labeled and improves sample efficiency; it does not change the reweighting procedure itself.

> Takeaway: HYRE is the unique coherent way to update ensemble weights from a loss function.
>
> The weight update (1) is axiomatically justified as generalized Bayesian inference (2), which concentrates on the best heads for each target distribution. This justification applies to non-differentiable metrics like 0-1 error. Empirically, HYRE is effective for preference-based shift when the ensemble spans the target, but not for structural covariate shift (Section 7).

## 4 When is Ensemble Reweighting Effective, and Why?

This section explores *the conditions under which ensemble reweighting is effective* through three illustrative examples: analyzing ensemble diversity through PCA, examining decision boundaries in classification, and comparing adaptation strategies.

**Ensemble diversity reflects task ambiguity.** We visualize how an ensemble's diversity reflects the axes of task ambiguity. We consider a synthetic regression task where the training data is sampled from a Gaussian Process (GP) prior. For target inputs $x_1, \ldots, x_M$, each ensemble member $f_k$ produces predictions $v_k \in \mathbb{R}^M$. We perform Principal Component Analysis (PCA) on the prediction matrix $V = (v_1, \ldots, v_K) \in \mathbb{R}^{M \times K}$ to yield components $u_1, \ldots, u_m \in \mathbb{R}^M$ that capture the main variations between ensemble members.

Using an ensemble of 100 models trained on 7 inputs and evaluated on 1000 test inputs, we visualize the first three principal components in Figure 3. Each component represents a distinct mode of variation while preserving smoothness and fit to training data. Like wavelets, these components are localized in input space

| Method | Energy | Kin8nm | CCPP |
|---|---|---|---|
| MC Dropout | 0.3033 | 0.6494 | 0.3761 |
| Vanilla | 0.1664 | 0.4514 | 0.2920 |
| + HYRE | 0.1572 (-0.0092) | 0.4498 (-0.0016) | 0.2902 (-0.0018) |
| Epinet | 0.1396 | 0.4823 | 0.3068 |
| + HYRE | 0.1345 (-0.0051) | 0.4814 (-0.0009) | 0.3036 (-0.0032) |
| Shared-Base | 0.1508 | 0.5316 | 0.2976 |
| + HYRE | 0.1431 (-0.0077) | 0.5314 (-0.0002) | 0.2955 (-0.0021) |

| Model | Helpful | Harmless |
|---|---|---|
| Fine-Tune (Helpful) | 73.03 | 32.59 |
| Fine-Tune (Harmless) | 32.06 | 73.30 |
| Pretrained RM | 68.01 | 52.16 |
| Ensemble | 66.34 | 50.90 |
| + HYRE (Helpful) | **68.44** | 51.21 |
| + HYRE (Harmless) | 64.24 | **57.66** |

Table 1: RMSE (lower is better) on test data with distribution shifts across three UCI datasets. We compare the performance of various ensemble architectures with test-time adaptation using HYRE. We find that **for all three ensemble architectures, HYRE is consistently able to adapt to the distribution shift between training and test data**.

Table 2: Helpful vs harmless tradeoff. To establish an upper bound on performance, we fine-tune the reward model on the helpful and harmless datasets separately. **Reweighting an ensemble model with HYRE allows us to flexibly trade off between the two desiderata**.

and form a basis for approximating the ensemble. See Section G for further analysis of PCA applied to ensemble predictions.

**Ensembles as diverse sharp decision boundaries.** We build on an alternative interpretation of the Bradley-Terry model, where the model can be seen as representing a population of deterministic decision-makers. For items $i$ and $j$ with parameters $\theta_i, \theta_j \in \mathbb{R}$, the preference probability under the Bradley-Terry model is:

$$P(i \succ j) = \frac{e^{\theta_i}}{e^{\theta_i} + e^{\theta_j}} = P\left(\theta_i + \epsilon_i > \theta_j + \epsilon_j\right), \tag{3}$$

where $\epsilon_i, \epsilon_j \sim \text{Gumbel}(0, 1)$. Rather than a single stochastic decision-maker, the model can be seen as representing a population of deterministic decision-makers. Each decision-maker is characterized by a pair $(\epsilon_i, \epsilon_j)$, and makes sharp choices based on which among $\theta_i + \epsilon_i$ and $\theta_j + \epsilon_j$ is larger. The model's probabilistic behavior emerges from averaging across this population. We do not claim that ensemble heads approximate Gumbel random variables; rather, the Gumbel interpretation of Bradley-Terry motivates a representation where distinct sharp decision-makers capture the population of annotators.

We hypothesize that diverse ensembles can learn such sharp decision boundaries from aggregate data across a population of annotators. To test this, we construct a synthetic preference learning task with conflicting labelers. We sample inputs $(x_1, x_2)$ from $[0, 1]^2$ and generate diverse linear decision boundaries $w_1 x_1 + w_2 x_2 > 0$, with $w_1, w_2 \sim N(0, 1)$. As shown in Figure 5, our ensemble quickly adapts to new decision boundaries, outperforming single models. The average ensemble prediction matches the "average" decision-maker, while individual members capture distinct boundaries. In particular, higher diversity coefficients for the prior network yields sharper boundaries per ensemble member. In Section 6, we show this enables rapid personalization in real-world preference tasks.

**HYRE outperforms fine-tuning in low-data regimes.** We compare HYRE to model fine-tuning on a synthetic binary classification task. The training set contains inputs from $[0, 1]^5$ labeled as 1 and inputs from $[-1, 0]^5$ labeled as 0. The target distribution is uniform over $[-1, 1]^5$ with a random linear decision boundary. Results in Figure 4 show that HYRE outperforms fine-tuning in the low-data regime, achieving high accuracy with few queries. Fine-tuning eventually surpasses reweighting with more data due to its higher capacity. This illustrates a bias-variance tradeoff: reweighting reduces variance by restricting solutions to the ensemble's span, providing an advantage with limited data. Additionally, HYRE requires only a single forward pass and negligible weight computation cost (1), making it especially suitable for large models and resource-constrained settings.

**Summary: when is HYRE effective?** HYRE is effective when the ensemble's functional diversity covers the target behavior, controlled by $K$ and the prior scale. It is less effective when the target lies outside this coverage, as in severe covariate shift settings (see WILDS results in Section 7), or when abundant adaptation data is available (where fine-tuning's higher capacity wins, as shown in Figure 4).

> **Key insight: Ensembles capture multiple valid interpretations**
>
> Ensembles naturally capture multiple valid interpretations of training data. Reweighting upweights the heads that best fit the adaptation set without retraining.

# 5    Related Work

**Ensemble methods.** A long-standing theme in machine learning is using ensembles to improve predictive performance and uncertainty estimates when different members make independent mistakes [39, 40]. Our approach builds on efficient ensemble methods with shared backbones [50] and extends prior work on dynamic ensemble weighting [34, 57], though these typically focus on differentiable loss-based objectives during training. In contrast, we perform adaptive reweighting at inference time using minimal labeled data from the target distribution, supporting non-differentiable evaluation metrics for effective alignment. Recent work in multi-objective optimization [26, 46, 47] uses Chebyshev scalarization with exponential weighting for balancing training objectives; while our weighting scheme is similar, we apply it to test-time adaptation rather than training-time optimization. This test-time mechanism can be interpreted as nonparametric meta-learning in hypothesis space, resembling prototypical networks [64, 71] in that adaptation is a closed-form computation over the support set with no gradient updates. Unlike MAML [18] and prototypical networks, HYRE requires no episodic meta-training with explicit task distributions; diversity comes from prior networks during standard training on a single dataset. While our use of multiple prediction heads may superficially resemble Mixture-of-Experts (MoE) [16, 33, 44, 59], the mechanisms differ: MoE uses per-token gating to route inputs to specialized sub-networks for compute efficiency, while HYRE applies a single global weighting across complete, standalone reward models to resolve preference ambiguity.

**Task underspecification and scalable alignment.** In many machine learning tasks, the training data fails to fully define desired model behavior [14, 23]. This challenge intensifies under limited data or distribution shifts, where multiple hypotheses remain consistent with observations. Reinforcement learning faces similar issues: reward specification is difficult in open-ended environments, and optimizing misspecified objectives can lead to unintended behaviors [21, 53, 63, 79]. Instead of fully defining a task upfront, one can collect human demonstrations or pairwise preferences, framing task specification as a cooperative game between agents and humans [24]. Reinforcement Learning from Human Feedback (RLHF) operationalizes this idea by using user preferences to guide post-training [11, 51, 55, 73], with some using ensembles [2, 12, 78]. Recent work on pluralistic alignment [65] uses explicit domain labels or per-user data to improve personalization [6, 10, 32, 45, 54]. However, these methods require explicit domain labels or per-user data. Specialized inference-time alignment approaches [17, 30, 35, 49] focus on token-level steering or reranking but assume a single fixed reward and cannot reconcile conflicting user preferences at test time. HYRE demonstrates that this additional information is not necessary during training: a diverse ensemble trained on aggregate data can capture ambiguity, which we can use to directly adapt to new users. Our experiments show this insight generalizes across several problem settings.

# 6    Experiments

We now empirically validate HYRE. We focus on three key questions: (1) Can HYRE effectively handle mild covariate shift? (2) Does HYRE scale to large models? (3) How robust and computationally efficient is HYRE? We describe the detailed setup for each experiment in the appendix.

## 6.1    Regression Data with Mild Covariate Shift

We evaluate HYRE on three UCI regression datasets [36]—Energy Efficiency, Kin8nm, and CCPP—using the protocol of Sharma et al. [58]: the top and bottom 5% of the data (sorted by mean input features) form an OOD target set, while the central 90% is split into train and validation sets. All methods employ 100 two-layer MLPs with 50 units each. As baselines, we consider a vanilla ensemble of independently trained models and MC Dropout [19]. We report the best-performing MC Dropout results across all architectures. Results in Table 1 demonstrate that uniform ensembles perform strongly in these OOD generalization settings and that HYRE consistently improves over the uniform ensemble.

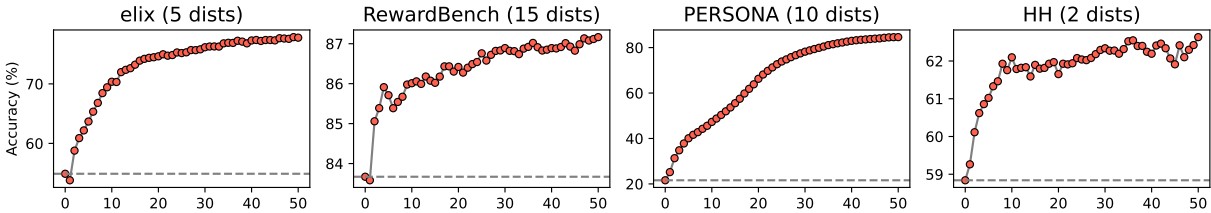

Figure 6: Average reward model accuracy as a function of adaptation set size. The dashed line shows the best available static 2B reward model for each dataset group. HYRE **consistently outperforms the state-of-the-art reward model with as few as 1-5 examples per distribution**.

| Model | Type | Overall | Chat | Chat Hard | Safety | Reasoning |
|---|---|---|---|---|---|---|
| Tulu-2-DPO-70B | DPO | 79.1 | 97.5 | 60.5 | 84.5 | 74.1 |
| Claude-3 Sonnet (June 2024) | Gen | 84.2 | 96.4 | 74.0 | 81.6 | 84.7 |
| GPT-4 (Aug 2024) | Gen | 86.7 | 96.1 | 76.1 | 88.1 | 86.6 |
| Gemini-1.5-Pro-0924 | Gen | 86.8 | 94.1 | 77.0 | 85.8 | 90.2 |
| RISE-Judge-Qwen2.5-7B | Gen | 88.2 | 92.2 | 76.5 | 88.0 | 96.1 |
| INF-ORM-Llama3.1-70B | Seq | 95.1 | 96.6 | 91.0 | 93.6 | 99.1 |
| GRM-Gemma2-2B | Seq | 88.4 | 93.0 | **77.2** | 92.2 | 91.2 |
| + Ours (uniform) | Seq | 87.1 | 96.4 | 73.1 | 87.4 | 89.8 |
| + Ours (N=1) | Seq + HYRE | 86.5 | 92.4 | 71.5 | 85.1 | 92.5 |
| + Ours (N=5) | Seq + HYRE | 88.5 | 95.0 | 72.5 | 90.3 | 93.1 |
| + Ours (N=10) | Seq + HYRE | **89.7** | **96.4** | 74.7 | **92.4** | **93.5** |
| + Ours (best weight oracle)* | Seq + Oracle | 93.1 | 98.3 | 83.4 | 96.7 | 94.9 |
| Skywork–Llama-3.1-8B | Seq | 94.0 | 94.7 | 88.6 | 92.7 | 96.7 |
| + Ours (uniform) | Seq | 94.0 | 95.0 | 87.2 | 93.0 | 96.8 |
| + Ours (N=1) | Seq + HYRE | 94.3 | 95.2 | 87.8 | 93.0 | 97.5 |
| + Ours (N=5) | Seq + HYRE | 94.7 | 95.5 | 88.6 | 93.2 | 97.8 |
| + Ours (N=10) | Seq + HYRE | **95.0** | **95.9** | **89.3** | **93.5** | **97.9** |
| + Ours (best weight oracle)* | Seq + Oracle | 97.2 | 99.2 | 93.0 | 96.5 | 98.8 |

 ∗ *Oracle* methods show an upper bound on performance, using the test set.

Table 3: Accuracy across tasks in RewardBench. We report overall performance and breakdowns by task category for all models. HYRE **improves upon the state-of-the-art models at the 2B and 8B parameter scales with as few as 1-5 labeled samples per distribution**.

## 6.2 Scalable Personalization of Preference Models

**Experimental setup.** We evaluate personalization using four sets of human preference benchmarks: Elix [61], RewardBench [41], PERSONA [9], and Anthropic HH [5]. Together, these benchmarks contain 32 datasets, each encoding a different aspect of human preferences. To train HYRE on preference data, we attach Shared-Base ensemble heads to a pretrained 2B reward model and fine-tune it on the UltraFeedback [13] dataset, a standard dataset for reward model training. We use two public fine-tuned Gemma 2B checkpoints that achieve state-of-the-art performance on RewardBench at the 2B-parameter scale, even outperforming GPT-4o [1]. Unless stated otherwise, we use a prior scale of 100.0 for all preference experiments, which we found to be robust across datasets. See Section D for our detailed setup and Section E for architecture details.

We first evaluate the effectiveness of HYRE in adapting our reward model ensemble to new distributions at test time, comparing its performance to that of the original reward model. As shown in Figure 6, a simple uniform ensemble initially underperforms the original model, indicating that naive ensembling alone cannot ensure broad generalization. Nevertheless, HYRE quickly surpasses this baseline with just a few labeled examples per distribution. We provide detailed dataset-level results in the appendix (Figure 9).

| Dataset | N=0 | N=1 | N=5 | N=10 | N=20 | N=40 | N=80 |
|---|---|---|---|---|---|---|---|
| **GPT-4o-mini N-Shot Prompting** | | | | | | | |
| donotanswer | 44.4 | 50.3 | 60.2 | 64.7 | 68.7 | 66.4 | 67.0 |
| refusals | 79.4 | 82.7 | 80.5 | 82.2 | 82.1 | 78.0 | 82.1 |
| **Llama-3.1-8B N-Shot Prompting** | | | | | | | |
| donotanswer | 46.6 | 52.8 | 59.6 | 63.4 | 41.2 | 62.6 | * |
| refusals | 61.6 | 82.4 | 80.0 | 72.2 | 44.4 | 79.0 | * |
| **Llama-3.1-8B + HYRE** | | | | | | | |
| donotanswer | 58.6 | 60.8 | 69.1 | **71.3** | | | |
| refusals | 88.9 | 90.0 | 94.0 | **95.2** | | | |

Table 4: Comparison with few-shot prompting on two datasets from RewardBench. (*) exceeds Together AI API token limit. We see a degradation in performance for both GPT-4o-mini and Llama-3.1-8B as we increase the number of examples, whereas HYRE consistently outperforms both across all sample sizes. HYRE **provides reliable test-time alignment, unlike few-shot prompting, which can degrade with too much context**.

| | donotanswer | xstest-sr | refusals |
|---|---|---|---|
| **HYRE w/ Cross-Entropy** | | | |
| N=0 | $58.60 \pm 4.93$ | $82.80 \pm 2.43$ | $88.90 \pm 3.25$ |
| N=1 | $62.53 \pm 4.00$ | $85.78 \pm 3.06$ | $92.49 \pm 3.28$ |
| N=5 | $62.57 \pm 1.88$ | $87.38 \pm 1.26$ | $93.17 \pm 2.19$ |
| N=10 | $62.25 \pm 1.71$ | $87.51 \pm 1.19$ | $93.21 \pm 1.77$ |
| **HYRE w/ Accuracy** | | | |
| N=0 | $58.60 \pm 4.93$ | $82.80 \pm 2.43$ | $88.90 \pm 3.25$ |
| N=1 | $60.81 \pm 5.29$ | $85.80 \pm 2.45$ | $90.00 \pm 3.63$ |
| N=5 | $69.12 \pm 5.81$ | $89.28 \pm 2.86$ | $94.00 \pm 2.45$ |
| N=10 | $\mathbf{71.32 \pm 6.33}$ | $\mathbf{90.32 \pm 3.20}$ | $\mathbf{95.20 \pm 2.32}$ |
| Oracle | $76.54 \pm 2.35$ | $90.32 \pm 1.91$ | $99.50 \pm 0.87$ |

Table 5: Cross-entropy ablation experiment. We report average and std of accuracy (%) with varying numbers of adaptation examples (N) on three datasets. **Using accuracy as the adaptation objective for HYRE significantly improves post-adaptation performance**.

| Method | Win Rate |
|---|---|
| Random | 50.0% |
| Uniform | 61.4% |
| HYRE (n=1) | 60.0% |
| HYRE (n=2) | 61.7% |
| HYRE (n=4) | 62.4% |
| HYRE (n=8) | 63.1% |
| HYRE (n=16) | 63.5% |

Table 6: Best-of-N win rates on Summarize from Feedback. HYRE **improves output quality with more examples.**

| Ratio | math-prm | xstest-sr |
|---|---|---|
| 0.0 : 1.0 | 72.57% | 88.33% |
| 0.1 : 0.9 | 98.94% | 86.64% |
| 0.2 : 0.8 | 96.73% | 88.15% |
| 0.5 : 0.5 | 98.52% | 87.22% |
| 0.8 : 0.2 | 99.38% | 86.66% |
| 0.9 : 0.1 | 99.52% | 85.86% |
| 1.0 : 0.0 | 99.72% | 84.18% |

Table 7: Accuracy with mixed adaptation data. HYRE **recovers 97% accuracy despite extreme imbalance.**

| Method | Acc (%) |
|---|---|
| Single Model | 59.03 |
| Entropy Weighted | 68.38 |
| Logit Ensemble [34] | 83.44 |
| Majority Vote | 83.71 |
| GEM [57, N=40] | 84.49 |
| GEM [57, Oracle†] | 89.51 |
| HYRE (N=1) | 83.88 |
| HYRE (N=5) | 85.73 |
| HYRE (N=10) | 86.26 |
| HYRE (N=20) | 87.11 |
| HYRE (N=40) | **87.74** |

Table 8: Comparison of ensemble aggregation methods. HYRE **outperforms all methods with only $N = 5$ examples.**

We compare HYRE against state-of-the-art reward models on the RewardBench leaderboard at both the 2B and 8B parameter scales. As shown in Table 3, HYRE—with only 1-5 labeled examples per distribution—exceeds the performance of many much larger reward models. We note that these reward models outperform strong generative reward models including Claude 3.5 Sonnet, GPT-4, and Gemini-1.5-Pro [1, 3, 68]. This suggests that inference-time alignment can be a strong alternative to simply scaling up reward models.

> **Takeaway: HYRE outperforms state-of-the-art reward models.**
>
> With just 1-5 labeled datapoints per distribution, HYRE outperforms state-of-the-art reward models on RewardBench at both the 2B and 8B parameter scales.

Beyond standard reward model accuracy, we validate that HYRE improves actual text generation quality using best-of-N sampling on the OpenAI Summarize from Feedback dataset [66], where multiple model-generated summaries are ranked by human Likert scores. As shown in Table 6, HYRE consistently outperforms the uniform ensemble baseline and scales with adaptation examples, translating reward model improvements into measurable downstream quality gains (McNemar's test p-values $10^{-16}$ to $10^{-3}$ across 4000+ comparisons).

## 6.3 Comparison with Alternative Adaptation Methods

**Fine-tuning on target data.** We compare HYRE against models fine-tuned on the helpful-base and harmless-base training sets in the Anthropic-HH dataset. Results in Table 2 indicate that while targeted fine-tuning models achieve higher performance in their respective target metrics, they significantly reduce performance in the other. In contrast, our HYRE-adapted ensemble not only increases performance across

each data distribution but also retains or slightly improves performance in the other split. We emphasize that we show fine-tuning performance only as a point of comparison; **fine-tuning a model for a target distribution is usually too computationally expensive to perform at inference time and is therefore not a practical solution for inference-time alignment**.

**Few-shot prompting.** We compare HYRE with few-shot prompting using GPT-4o-mini on two datasets from RewardBench. As shown in Table 4, HYRE consistently outperforms GPT-4o-mini across all sample sizes. Even in the zero-shot setting, specialized reward models like HYRE achieve higher performance than general-purpose language models like GPT-4o-mini. Notably, we observe that too many few-shot examples can actually harm performance, as seen with GPT-4o-mini's performance drop after N=10 for donotanswer and N=20 for refusals. These results demonstrate that specialized reward models with inference-time adaptation can more efficiently leverage few-shot examples than general-purpose language models.

### 6.4 Analysis and Ablation Studies

We analyze the computational cost of HYRE and ablate its key design choices: the reweighting objective, softmax temperature, ensemble size, and ensemble diversity.

**Computational overhead.** HYRE uses a single pre-trained backbone with $K$ small prediction heads, ensuring minimal parameter overhead. In our reward model experiments, 100 ensemble heads (550K parameters) add only 0.03% to the Gemma-2B backbone. At inference time, reweighting requires a single forward pass through the backbone and heads, with negligible weight computation. The total cost increase in time and memory is less than 1%, making the approach practical for deployment in production systems where fine-tuning would be prohibitively expensive.

**Ablation on reweighting objective.** We compare accuracy-based reweighting against binary cross-entropy loss. Cross-entropy significantly degrades performance across three RewardBench datasets (Table 5) because it is unbounded and sensitive to outliers, causing rapid overfitting to individual heads. This validates our use of generalized Bayesian inference with non-differentiable metrics rather than likelihood-based updates.

**Temperature for ensemble reweighting.** Generalizing (1) with a temperature parameter, $w_k \propto \exp(-\mathcal{L}_k/\tau)$, controls the sharpness of head selection (Table 11). At low temperatures ($\tau \leq 0.1$), weights collapse to a single head and additional examples provide little benefit. At high temperatures ($\tau \geq 5$), weights flatten toward uniform, also reducing gains. The default $\tau=1$ (i.e., (1) as stated) achieves the best accuracy at $N \geq 5$; we use this throughout.

**Ensemble size.** We subsample $K \in \{5, 10, 25, 50, 100\}$ heads to study the effect of ensemble size (Table 13). We find diminishing returns with $K$: most of the improvement comes from $K=5$ to $K=25$, with smaller gains beyond $K \approx 50$. The benefit of adaptation is consistent across all ensemble sizes.

**Effective ensemble size after adaptation.** We measure the effective number of contributing heads $K_{\text{eff}}$ via the entropy of the weight vector (Table 12). When heads disagree more, adaptation concentrates weight on fewer reliable heads. We find that subsets with low $K_{\text{eff}}$ show the largest accuracy gains from reweighting, confirming that adaptation does the most work where it is most needed.

**Robustness to skewed adaptation data.** We test HYRE on non-i.i.d. scenarios by mixing two Reward-Bench datasets (math-prm and xstest-should-respond) at different ratios. We train on these different mixtures, then evaluate the resulting ensemble on each distribution separately. Even with highly skewed mixtures (e.g., only 10% from dataset A), HYRE recovers 97% of the accuracy gains compared to adapting exclusively on dataset A (Table 7). This robustness to noisy or mixed adaptation signals is crucial for real-world deployment, where users may provide inconsistent or multi-domain feedback.

**Comparison with alternative reweighting methods.** We compare HYRE against existing ensemble weighting approaches: uniform weighting [34], confidence weighting, majority voting, and convex optimization [57]. HYRE consistently outperforms all prior methods with just 1-5 examples per distribution (Table 8), demonstrating that our update is more sample-efficient than existing ad-hoc weighting heuristics.

**Source of improvements.** We summarize the evidence for the source of HYRE's gains. **(1)** The uniform ensemble does not improve over the base model (Table 3, uniform rows), indicating that the gains do not come

from ensembling alone. **(2)** HYRE outperforms few-shot prompting given the same $N$ examples (Table 4), showing that reweighting is more effective than in-context learning in this setting. **(3)** HYRE outperforms fine-tuning in the low-data regime (Figure 4, Table 2), showing that reweighting is more effective than parameter updates when adaptation data is scarce. **(4)** HYRE outperforms all alternative ensemble weighting methods at the same $N$ (Table 8), indicating that the Bayesian update itself drives the gains.

> **Practical advantages of HYRE**
>
> HYRE shifts adaptation from parameter updates to a lightweight weight update over ensemble heads. In our 2B reward models, this adds only 0.03% parameters and less than 1% computational overhead, while requiring just one forward pass rather than hundreds of gradient steps.

## 7 Discussion

Our results show that efficient ensemble architectures can provide a practical mechanism for inference-time personalization in large models. The proposed method, HYRE, is well suited to settings where fine-tuning is impractical, including regulatory-frozen medical models that would require recertification after any parameter change, real-time personalization for millions of users under strict latency constraints, and live content moderation where norms drift faster than batch retraining cycles. By attaching lightweight ensemble "heads" to a shared backbone, we can capture multiple plausible interpretations of the training distribution at negligible extra cost. Then, using just a handful of target-domain examples, the reweighting step can identify the heads whose behavior best matches the new task. A natural next step is to close the loop by pairing our approach with a parameterization of the reward model that allows for direct behavior adjustments [55].

Our method currently relies on a small batch of labeled examples from the target distribution and does not address single-sample or online streaming adaptation. The reweighting also assumes that the ensemble's functional diversity covers the new domain's core behaviors. In the extreme case where all ensemble members collapse to the same function, any weighting reduces to that function and adaptation has no effect. We do not have general guarantees against collapse in neural network ensembles; instead, we rely on prior networks [50], which are empirically effective for maintaining diversity. Extending our framework to dynamically expand or augment the ensemble as new tasks emerge is an exciting direction. Nevertheless, our results show that lightweight ensembles with inference-time reweighting offer a practical approach to personalizing large models at inference time.

**When does HYRE help?** HYRE shows the strongest improvements on personalization tasks, where ensemble diversity can capture different valid interpretations of the same underlying task. On four additional WILDS datasets (CivilComments, Amazon, FMoW, iWildCam), reweighting via HYRE did not improve over the uniform ensemble (Table 15). We hypothesize that these shifts are structural rather than preference-based, so the target behavior may lie outside the functional diversity captured by the ensemble. These results suggest that HYRE is most effective when ambiguity arises from preference disagreement rather than structural domain shift.

## Acknowledgments

We thank Benjamin Van Roy for helpful input on formalizing our learning objective. We thank Collin Burns, Ruiqi Zhong, Niki Howe, members of the IRIS lab, and the TMLR reviewers for discussions and feedback on earlier drafts of this work. This work was partly supported by OpenAI, KFAS, NSF CAREER award 2237693, and Schmidt Sciences.

## Ethics Statement

This work adheres to the TMLR Ethics Guidelines. Our research focuses on improving preference learning methods, which have positive implications for AI alignment and human-AI collaboration. The methods developed could potentially be misused to optimize for harmful content, but the same risk exists with any preference learning approach. Our contribution lies in making such optimization more efficient rather than enabling fundamentally new capabilities. We encourage responsible deployment of these techniques with appropriate safety measures and content moderation systems. Inference-time adaptation raises specific governance concerns: because HYRE adapts to user-provided examples, adversarial adaptation data could steer the model toward harmful preferences. Because the weights and adaptation examples are explicit, deployments can log and audit adaptation behavior. Mitigations include filtering adaptation data against known harmful content and constraining permissible feedback types.

## Reproducibility Statement

Complete experimental details, including hyperparameters, architectures, and evaluation protocols, are provided in the main text and appendix. All pretrained models and datasets used in our experiments are publicly available.

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

| Method | N=0 | N=1 | N=5 | N=10 | N=20 | N=40 |
|---|---|---|---|---|---|---|
| HYRE + Random | 84.40 | 85.33 | 86.97 | 87.34 | 88.01 | 88.83 |
| HYRE + Entropy | 84.40 | 84.25 | 86.73 | 87.54 | 88.60 | 89.76 |
| HYRE + BALD | 84.40 | 84.28 | 87.13 | 87.78 | 88.60 | 88.99 |

Table 9: Accuracies on RewardBench with different datapoint selection strategies. While active sampling methods perform slightly better, **even random sampling consistently improves performance with the** HYRE **reweighting process**.

| Algorithm | DL | Test Acc |
|---|---|---|
| IRM | O | 64.2 (8.1) |
| CORAL | O | 59.5 (7.7) |
| Group DRO | O | 68.4 (7.3) |
| Fish | O | 74.7 (7.1) |
| LISA | O | 77.1 (6.9) |
| ERM | X | 70.3 (6.4) |
| Evading | X | 73.6 (3.7) |
| Ensemble | X | 71.5 (3.4) |
| Ensemble + HYRE | X | **75.2** (5.3) |

Table 10: Test set accuracy on Camelyon17. HYRE achieves competitive performance without using domain labels (DL).

## A Additional Experiments

**Effect of sampling strategy.** In Table 9, we compare the performance of different active learning criteria for selecting adaptation data points. We consider random sampling, BALD, and entropy, measuring their performance over 0 to 40 target examples. Across the acquisition of 40 examples, active learning methods (BALD and entropy) demonstrated slightly better performance compared to random sampling. Even random sampling consistently improves performance, indicating that HYRE remains effective when adaptation data is collected before inference.

**WILDS experiments.** We evaluate a trained Shared-Base ensemble, both with and without HYRE on the WILDS-Camelyon17 dataset [38], comparing against several representative methods for OOD generalization from the official WILDS benchmark. As shown in Table 10, test-time adaptation with HYRE consistently outperforms other methods that do not use domain labels and remains competitive with LISA [76], a strong method that leverages domain labels for targeted data augmentation. We also test Shared-Base ensembles on four additional WILDS datasets (CivilComments, Amazon, FMoW, iWildCam), but did not observe further improvements from ensemble reweighting via HYRE, as detailed in Table 15. Nonetheless, training a diverse ensemble consistently improved OOD generalization in these datasets.

We attribute the limited benefit of ensemble reweighting in these cases to some natural distribution shifts behaving similarly to in-distribution data in terms of task underspecification. For further discussion on the conditions that can make a single model outperform the ensemble, see Section 4.

We further compare the performance of HYRE with few-shot fine-tuning with the same amount of adaptation data. We evaluate both HYRE and fine-tuning with $\{4, 8, 16, 32\}$ datapoints from the OOD test set. Our results in Figure 7 show that ensemble reweighting outperforms fine-tuning in the low-data regime (4 and 8 examples), while fine-tuning eventually surpasses ensemble reweighting as more adaptation data becomes available.

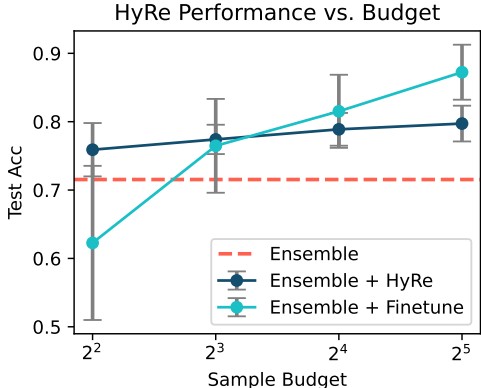

Figure 7: Comparison of HᴙRᴇ and few-shot fine-tuning on the Camelyon17 OOD test set. HᴙRᴇ outperforms fine-tuning in the low-data regime despite requiring significantly less computational cost.

Table 11: **Temperature ablation.** Average accuracy (%) across 10 RewardBench subsets for varying softmax temperature $\tau$ in the weight update $w_k \propto \exp(-\mathcal{L}_k/\tau)$. $\tau{=}1$ (the default) performs best at higher adaptation budgets. Low temperatures ($\tau \leq 0.1$) collapse to near-hard selection and negate the benefit of larger $N$; high temperatures ($\tau \geq 5$) flatten weights toward uniform. Error bars denote the average per-dataset 95% CI. **Bold**: best value per column (including ties). Blue: within the CI of the column best. Shaded row : default setting ($\tau{=}1$).

| $\tau$ | $N{=}1$ | $N{=}2$ | $N{=}5$ | $N{=}10$ | $N{=}20$ |
|---|---|---|---|---|---|
| 0.01 | $\mathbf{92.88}_{\pm 0.33}$ | $92.74_{\pm 0.32}$ | $93.06_{\pm 0.35}$ | $92.87_{\pm 0.40}$ | $92.90_{\pm 0.40}$ |
| 0.05 | $92.86_{\pm 0.32}$ | $92.68_{\pm 0.32}$ | $93.05_{\pm 0.35}$ | $92.78_{\pm 0.39}$ | $92.87_{\pm 0.39}$ |
| 0.10 | $92.85_{\pm 0.31}$ | $92.69_{\pm 0.32}$ | $92.97_{\pm 0.35}$ | $92.96_{\pm 0.39}$ | $93.09_{\pm 0.37}$ |
| 0.50 | $92.79_{\pm 0.31}$ | $92.76_{\pm 0.32}$ | $\mathbf{93.12}_{\pm 0.33}$ | $93.42_{\pm 0.34}$ | $93.54_{\pm 0.34}$ |
| 1.00 | $92.73_{\pm 0.32}$ | $\mathbf{92.90}_{\pm 0.33}$ | $\mathbf{93.12}_{\pm 0.32}$ | $\mathbf{93.44}_{\pm 0.32}$ | $\mathbf{93.72}_{\pm 0.32}$ |
| 2.00 | $92.56_{\pm 0.31}$ | $92.73_{\pm 0.31}$ | $92.97_{\pm 0.31}$ | $93.18_{\pm 0.31}$ | $93.50_{\pm 0.30}$ |
| 5.00 | $92.58_{\pm 0.32}$ | $92.68_{\pm 0.32}$ | $92.85_{\pm 0.32}$ | $92.99_{\pm 0.31}$ | $93.24_{\pm 0.31}$ |
| 10.0 | $92.40_{\pm 0.32}$ | $92.47_{\pm 0.32}$ | $92.56_{\pm 0.32}$ | $92.70_{\pm 0.32}$ | $92.85_{\pm 0.32}$ |

## B  Ablation Studies

We present two ablations and one analysis using a 100-head Gemma-2B shared-base ensemble trained on UltraFeedback and evaluated on 21 RewardBench subsets. All results report accuracy averaged over 200 bootstrap repeats of the adaptation/evaluation split. Error bars report the 95% CI for each dataset (via SEM over bootstrap repeats), averaged across the 21 subsets.

**Temperature (Table 11).** The softmax temperature $\tau$ in the weight update $w_k \propto \exp(-\mathcal{L}_k/\tau)$ controls the sharpness of head selection. At low temperatures ($\tau \leq 0.1$), weights concentrate on a single head and the $N$-scaling benefit largely disappears (+0.02 percentage points (pp) from $N{=}1$ to $N{=}20$ at $\tau{=}0.01$ vs. +1.0 pp at $\tau{=}1.0$). At high temperatures ($\tau \geq 5$), weights flatten toward uniform, also reducing gains. The default $\tau{=}1$ achieves the best accuracy at $N \geq 5$.

**Effective ensemble size (Table 12).** To quantify how many heads contribute meaningfully after adaptation, we compute $K_{\text{eff}} = \exp(-\sum_k w_k \log w_k)$, the perplexity of the weight distribution. This ranges from 1 (all weight on a single head) to $K$ (uniform weights). Across subsets, $K_{\text{eff}}$ averages 43.1 out of 100 heads, indicating that adaptation concentrates weight on roughly 43% of heads while retaining soft coverage over the rest. The pairwise disagreement rate (Disagree) is negatively correlated with $K_{\text{eff}}$: when heads disagree more, adaptation concentrates weight on fewer reliable heads, as reflected in max $w_k$. LLMBar Manual has the highest single-head weight (0.242) and lowest $K_{\text{eff}}$ (21.5); HumanEval C++ has max weight 0.015, nearly uniform across all 100 heads. By group, $K_{\text{eff}}$ is highest for code tasks (HumanEval, 61–76) where correctness

Table 12: **Ensemble diversity.** For each RewardBench subset, we report the effective number of heads (perplexity of the weight distribution) after $N=20$ adaptation steps, the maximum head weight, the pairwise disagreement rate between individual heads and the ensemble (Disagree), and the accuracy spread (max $-$ min per-head accuracy). On average, $K_{\text{eff}} \approx 43$ of 100 heads carry meaningful weight after adaptation.

| Dataset | $K_{\text{eff}}$ | Max $w_k$ | Disagree | Acc Spread |
|---|---|---|---|---|
| *AlpacaEval* | | | | |
|     Easy | 34.0 | 0.112 | 0.173 | 0.670 |
|     Hard | 33.5 | 0.111 | 0.199 | 0.632 |
|     Length | 35.9 | 0.091 | 0.179 | 0.642 |
| DoNotAnswer | 26.1 | 0.211 | 0.225 | 0.552 |
| *HumanEval* | | | | |
|     C++ | 75.8 | 0.015 | 0.136 | 0.793 |
|     Go | 67.6 | 0.019 | 0.159 | 0.884 |
|     Java | 73.8 | 0.016 | 0.130 | 0.842 |
|     JS | 68.1 | 0.020 | 0.157 | 0.805 |
|     Python | 61.1 | 0.024 | 0.164 | 0.829 |
|     Rust | 65.5 | 0.019 | 0.170 | 0.768 |
| *LLMBar* | | | | |
|     Adver-GPTInst | 28.2 | 0.113 | 0.229 | 0.707 |
|     Adver-GPTOut | 25.6 | 0.179 | 0.260 | 0.532 |
|     Adver-Manual | 21.5 | 0.242 | 0.278 | 0.500 |
|     Adver-Neighbor | 24.2 | 0.167 | 0.242 | 0.619 |
|     Natural | 26.3 | 0.178 | 0.206 | 0.490 |
| Math-PRM | 37.9 | 0.068 | 0.162 | 0.966 |
| MT-Bench Medium | 48.8 | 0.037 | 0.145 | 0.525 |
| *Refusals* | | | | |
|     Dangerous | 39.1 | 0.098 | 0.184 | 0.930 |
|     Offensive | 57.5 | 0.031 | 0.146 | 1.000 |
| *XSTest* | | | | |
|     Should-Refuse | 31.4 | 0.102 | 0.201 | 0.948 |
|     Should-Respond | 24.1 | 0.156 | 0.219 | 0.748 |
| **Average** | **43.1** | 0.096 | 0.189 | 0.732 |

Table 13: **Ensemble size ablation.** Average accuracy (%) across 10 RewardBench subsets for varying numbers of ensemble heads $K$. For $K < 100$, we randomly subsample heads and evaluate without retraining (20 subsamples $\times$ 50 repeats each). Performance increases logarithmically with $K$, with diminishing returns beyond $K \approx 50$. Error bars denote the average per-dataset 95% CI.

| $K$ | $N=0$ | $N=1$ | $N=2$ | $N=5$ | $N=10$ | $N=20$ |
|---|---|---|---|---|---|---|
| 5 | $89.94_{\pm 1.88}$ | $90.50_{\pm 1.56}$ | $90.67_{\pm 1.46}$ | $90.93_{\pm 1.36}$ | $90.95_{\pm 1.36}$ | $90.89_{\pm 1.39}$ |
| 10 | $91.13_{\pm 1.06}$ | $91.62_{\pm 0.82}$ | $91.84_{\pm 0.74}$ | $92.10_{\pm 0.68}$ | $92.23_{\pm 0.68}$ | $92.08_{\pm 0.73}$ |
| 25 | $92.20_{\pm 0.55}$ | $92.47_{\pm 0.49}$ | $92.58_{\pm 0.46}$ | $92.79_{\pm 0.42}$ | $93.02_{\pm 0.43}$ | $93.03_{\pm 0.43}$ |
| 50 | $92.33_{\pm 0.40}$ | $92.60_{\pm 0.33}$ | $92.72_{\pm 0.30}$ | $92.96_{\pm 0.28}$ | $93.26_{\pm 0.28}$ | $93.41_{\pm 0.29}$ |
| 100 | $92.40_{\pm 0.14}$ | $92.70_{\pm 0.13}$ | $92.85_{\pm 0.14}$ | $93.08_{\pm 0.14}$ | $93.44_{\pm 0.13}$ | $93.67_{\pm 0.14}$ |

is objective and heads agree, lowest for adversarial preference tasks (LLMBar, 22–28) where the algorithm concentrates on a small specialist subset, and intermediate for safety (24–58) and general preference (34–49) tasks.

**Number of heads (Table 13).** We subsample $K \in \{5, 10, 25, 50, 100\}$ heads (20 random subsamples each, 50 repeats per subsample) to study the effect of ensemble size. Performance shows diminishing returns: most

of the improvement comes from $K=5$ to $K=25$, with smaller gains beyond $K \approx 50$. Per-dataset variance also decreases with $K$, reflecting both reduced head-subsampling noise and the stability of larger ensembles. The benefit of adaptation is consistent across all ensemble sizes, though slightly attenuated at small $K$ where the ensemble's functional diversity is more limited.

## C  Active Learning Details

We also consider an active learning setup in which the $N$ datapoints to label for HYRE are chosen at test time from a larger unlabeled pool of data. Rather than choosing all datapoints at once, we choose one datapoint at the time based on one of the following three criteria:

- **Entropy** (classification): $H\left(\sum_{h=1}^{H} w_h f_h(x)\right)$. This criterion selects datapoints where the weighted ensemble is most uncertain, promoting the exploration of ambiguous regions.

- **BALD** (classification): $H\left(\sum_{i=1}^{H} w_i f_i(x)\right) - \sum_{i=1}^{H} w_i H(f_i(x))$. BALD considers both ensemble uncertainty and disagreement among members, balancing exploration and exploitation [20, 27].

- **Variance** (regression): $\sum_{i=1}^{H} w_i(f_i(x) - \bar{f}(x))^2$, where $\bar{f}(x) = \sum_{i=1}^{H} w_i f_i(x)$. This criterion focuses on points where ensemble predictions have the highest variance, which is a good indicator of uncertainty in regression tasks.

Each of these criteria can be computed quickly. Because the belief states $w$ has a closed-form update that can be computed very quickly, we can efficiently recompute the next best data point after each active label query.

We note that the first criterion (Entropy) does not distinguish between so-called aleatoric uncertainty and epistemic uncertainty. Therefore, this criterion is susceptible to the "noisy TV problem", where an agent fixates on a source of uncertainty that cannot be resolved [8, 42]. In practice, we find that HYRE is robust to the choice of active learning criterion, and even random selection is effective at adapting to the target distribution.

## D  Experimental Details

Unless specified otherwise, we use the following configuration for the ensemble networks. We use an ensemble of 100 models. The learnable and prior networks are each a one-hidden-layer MLP with 128 units. For the epinet, the epistemic index is 10-dimensional. For ensemble reweighting via HYRE, we use 32 examples from the target dataset, actively queried based on the BALD (classification) or Variance (regression) criterion. We found that final performance is not very sensitive to the choice of active learning criterion, and even random sampling resulted in consistent benefits.

**WILDS.** We closely follow the reference WILDS implementation for each dataset [38], including the choice of backbone, learning rate, and weight decay. We briefly describe the baseline methods used in our experiments:

- **CORrelation ALignment** [67, CORAL]: CORAL is an unsupervised domain adaptation method that aligns the second-order statistics (covariances) of source and target feature distributions.

- **Invariant Risk Minimization** [4, IRM]: IRM aims to learn data representations that capture invariant correlations across multiple training distributions.

- **Group Distributionally Robust Optimization** [56, Group DRO]: Group DRO seeks to minimize the worst-case training loss over predefined groups within the data.

- **Fish** [60]: Fish is a domain generalization technique that approximates inter-domain gradient matching by maximizing the inner product between gradients from different domains.

- **LISA** [76]: LISA builds on MixUp and selectively interpolates data samples to achieve domain invariance.

| Model | Type | Overall | Chat | Chat Hard | Safety | Reasoning |
|---|---|---|---|---|---|---|
| Mixtral-8x7B-Instruct-v0.1 | DPO | 77.6 | 95.0 | 64.0 | 72.6 | 78.7 |
| LLaMA-3-Tulu-2-DPO-70B | DPO | 77.2 | 96.4 | 57.5 | 74.9 | 80.2 |
| Tulu-2-DPO-13B | DPO | 76.7 | 95.8 | 58.3 | 79.5 | 73.2 |
| Tulu-2-DPO-70B | DPO | 79.1 | 97.5 | 60.5 | 84.5 | 74.1 |
| StableLM-2-12B-Chat | DPO | 79.9 | 96.6 | 55.5 | 78.1 | 89.4 |
| Claude-3 Sonnet (June 2024) | Gen | 84.2 | 96.4 | 74.0 | 81.6 | 84.7 |
| GPT-4 (May 2024) | Gen | 84.6 | 96.6 | 70.4 | 86.5 | 84.9 |
| GPT-4 (Aug 2024) | Gen | 86.7 | 96.1 | 76.1 | 88.1 | 86.6 |
| Gemini-1.5-Pro-0924 | Gen | 86.8 | 94.1 | 77.0 | 85.8 | 90.2 |
| Skywork-Reward-Gemma-2-27B | Seq | 94.3 | 96.1 | 89.9 | 93.0 | 98.1 |
| INF-ORM-Llama3.1-70B | Seq | 95.1 | 96.6 | 91.0 | 93.6 | 99.1 |
| GRM-Gemma-2B | Seq | 84.5 | 89.4 | **75.2** | 84.5 | 88.8 |
| + Ours (uniform) | Seq | 84.5 | 88.6 | 72.9 | 83.7 | 89.8 |
| + Ours (N=1) | Seq + HYRE | 85.3 | 88.5 | 72.7 | 85.5 | 91.4 |
| + Ours (N=5) | Seq + HYRE | 86.4 | 90.3 | 72.6 | 89.1 | 91.4 |
| + Ours (N=10) | Seq + HYRE | **87.2** | **90.4** | 72.5 | **90.0** | **92.3** |
| + Ours (best head oracle)[*] | Seq + Oracle | 88.6 | 91.1 | 78.1 | 91.9 | 92.3 |
| + Ours (best weight oracle)[*] | Seq + Oracle | 90.0 | 92.3 | 81.8 | 92.5 | 93.1 |
| GRM-Gemma2-2B | Seq | 88.4 | 93.0 | **77.2** | 92.2 | 91.2 |
| + Ours (uniform) | Seq | 87.1 | 96.4 | 73.1 | 87.4 | 89.8 |
| + Ours (N=1) | Seq + HYRE | 86.5 | 92.4 | 71.5 | 85.1 | 92.5 |
| + Ours (N=5) | Seq + HYRE | 88.5 | 95.0 | 72.5 | 90.3 | 93.1 |
| + Ours (N=10) | Seq + HYRE | **89.7** | **96.4** | 74.7 | **92.4** | **93.5** |
| + Ours (best head oracle)[*] | Seq + Oracle | 91.8 | 97.2 | 80.0 | 96.2 | 94.2 |
| + Ours (best weight oracle)[*] | Seq + Oracle | 93.1 | 98.3 | 83.4 | 96.7 | 94.9 |
| Skywork–Llama-3.1-8B | Seq | 94.0 | 94.7 | 88.6 | 92.7 | 96.7 |
| + Ours (uniform) | Seq | 94.0 | 95.0 | 87.2 | 93.0 | 96.8 |
| + Ours (N=1) | Seq + HYRE | 94.3 | 95.2 | 87.8 | 93.0 | 97.5 |
| + Ours (N=5) | Seq + HYRE | 94.7 | 95.5 | 88.6 | 93.2 | 97.8 |
| + Ours (N=10) | Seq + HYRE | **95.0** | **95.9** | **89.3** | **93.5** | **97.9** |
| + Ours (best head oracle)[*] | Seq + Oracle | 96.4 | 98.3 | 91.2 | 95.7 | 98.4 |
| + Ours (best weight oracle)[*] | Seq + Oracle | 97.2 | 99.2 | 93.0 | 96.5 | 98.8 |

  ∗ *Oracle* methods show an upper bound on performance, using the test set.

Table 14: Accuracy across tasks in RewardBench (full version of Table 3). We report overall performance and breakdowns by task category for all models. HYRE **improves upon the state-of-the-art models at the 2B and 8B parameter scales with as few as 1-5 labeled samples per distribution**.

**LLM Preference Learning** We finetune three reward model checkpoints [75]:

- https://huggingface.co/Ray2333/GRM-Gemma-2B-rewardmodel-ft

- https://huggingface.co/Ray2333/GRM-Gemma2-2B-rewardmodel-ft

- https://huggingface.co/Skywork/Skywork-Reward-Llama-3.1-8B-v0.2

Our ensemble architecture uses these networks as the backbone, and small MLPs for the learnable and prior networks which take the backbone's final embedding as input. We use the TRL codebase for reward model training [72]. We train with bfloat16 mixed precision. We use a learning rate of 0.0001, no weight decay, a batch size of 16, and train for 5000 steps. We consider four collections of preference datasets:

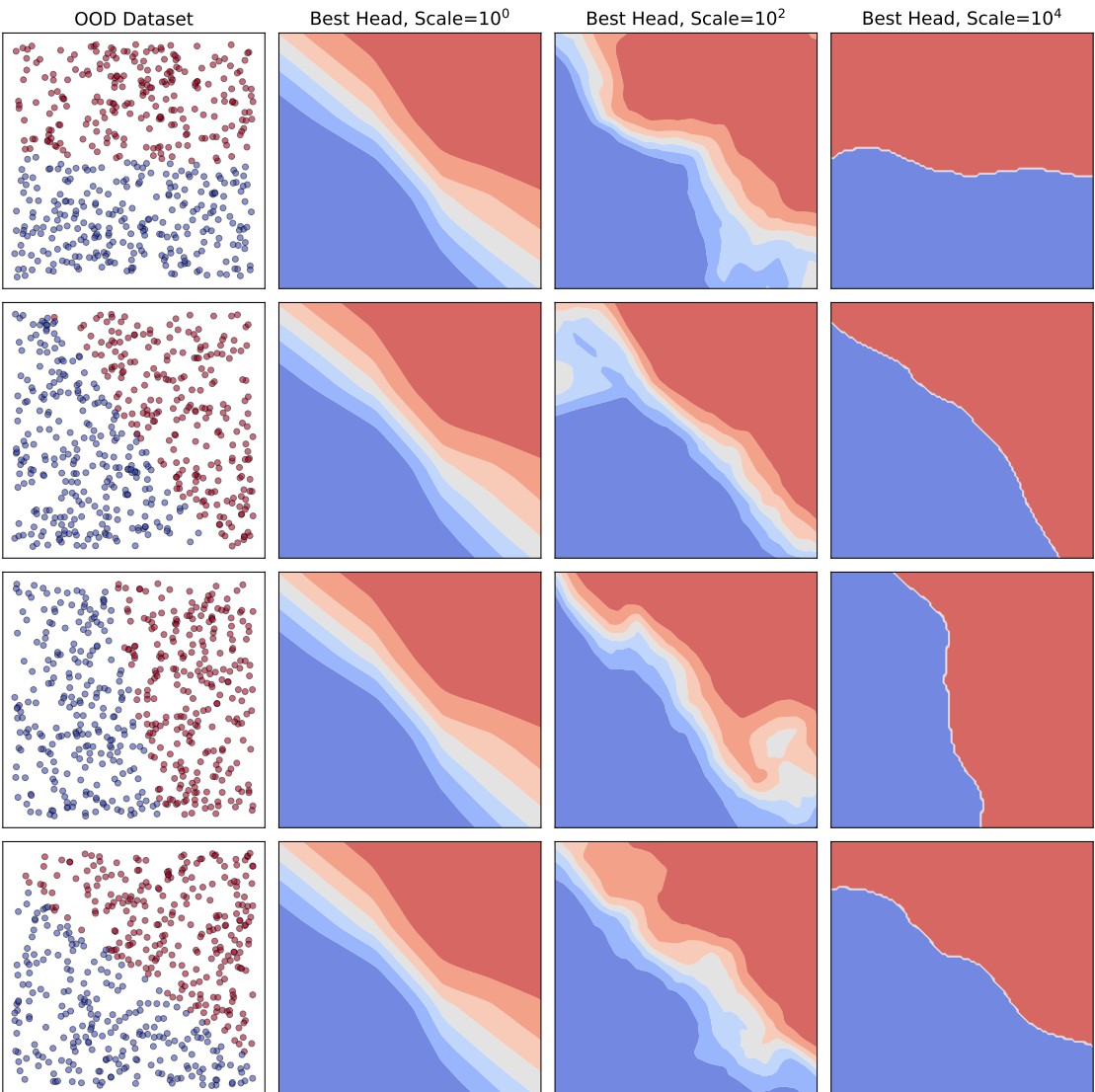

Figure 8: Additional visualizations for the toy conflicting classification example. Increasing the scale hyperparameter results produces heads with sharper decision boundaries.

- **Elix** [61] is inspired by the "Explain like I'm 5" subreddit. It consists of questions answered at five educational levels: elementary, middle, high, college, and expert. Preference pairs are created by scoring how different pairs of GPT-4 generated responses meet the expected comprehension at each level.

- **RewardBench** [41] is a suite of 27 preference datasets designed to test reward models on a broad spectrum of tasks, including chat quality, safety, reasoning, coding, and refusal handling. In our aggregate results Figure 6, we drop datasets with less than 100 examples. In our RewardBench experiments Table 3, we use all datasets to ensure a fair comparison with existing methods.

- **PERSONA** [9] contains preference data derived from a collection of synthetic personas with diverse demographic attributes and values. We sample 10 personas and treat each as a target distribution. Further details are in Section H.

- **Anthropic HH** [5] contains human-labeled preferences focused on helpfulness and harmlessness. We use the helpfulness-base and harmlessness-base splits as evaluation distributions to measure the tradeoff between the two objectives.

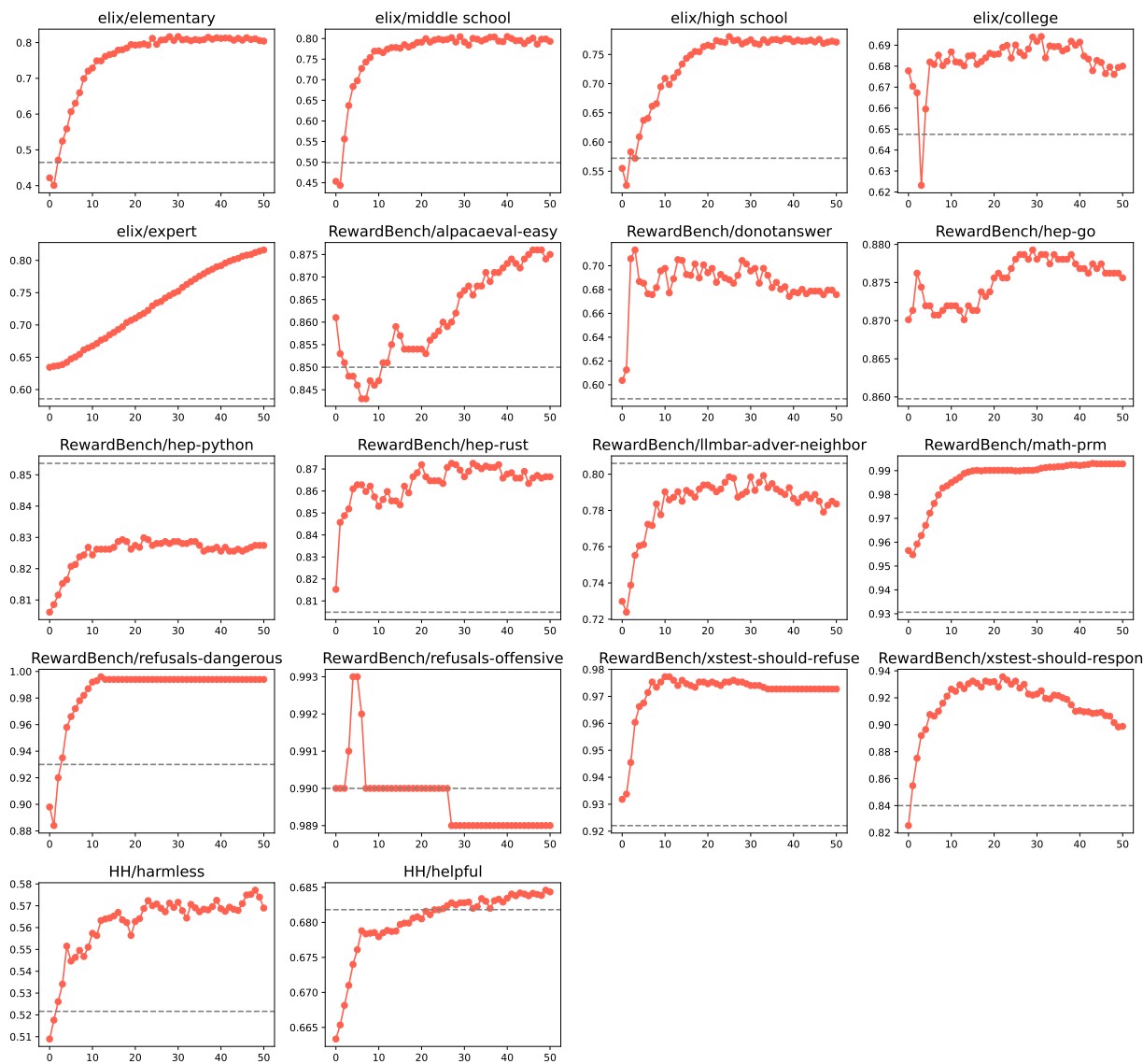

Figure 9: Detailed results for the personalizing preference reward models experiment in Figure 6. Target dataset accuracy (y-axis) after observing different numbers of adaptation samples (x-axis). The dashed line represents the performance of the pretrained reward model.

For the few-shot prompting experiments, we use GPT-4o-mini. For each number of "shots" $N \in \{0, 1, 5, 10, 20, 40, 80\}$, we sample 1000 examples from the target distribution and use them to prompt GPT-4o-mini.

# E    Diverse Ensemble Architectures

We describe the diverse ensemble architectures used in our experiments. Each architecture is designed to parameterize an ensemble of $H$ models, whose outputs are later combined to form an ensemble prediction. The key goal of these architectures is to produce diverse predictions across the ensemble at a low computational cost.

| Algorithm | DL | CivilComments
Worst-Group Acc | Amazon
10% Acc | FMoW
Worst-Reg Acc | iWildCam
Macro F1 |
|---|---|---|---|---|---|
| IRM | O | 66.3 (2.1) | 52.4 (0.8) | 32.8 (2.09) | 15.1 (4.9) |
| IRMX | O | 73.4 (1.4) | - | 33.7 (0.95) | 26.7 (1.1) |
| IRMX (PAIR) | O | 74.2 (1.4) | - | 35.4 (1.3) | 27.9 (0.9) |
| CORAL | O | 65.6 (1.3) | 52.9 (0.8) | 32.8 (0.66) | 32.7 (0.2) |
| Group DRO | O | 70.0 (2.0) | 53.3 (0.0) | 31.1 (1.66) | 23.8 (2.0) |
| DFR | O | 72.5 (0.9) | - | 42.8 (0.42) | - |
| Fish | O | 75.3 (0.6) | 53.3 (0.0) | 34.6 (0.18) | 22.0 (1.8) |
| LISA | O | 72.9 (1.0) | 54.7 (0.0) | 35.5 (0.81) | - |
| ERM | X | 56.0 (3.6) | 53.8 (0.8) | 31.3 (0.17) | 30.8 (1.3) |
| Shared-Base | X | 58.1 (2.2) | 54.2 (0.6) | 32.8 (0.4) | 30.9 (0.8) |
| Shared-Base + HYRE | X | 58.1 (0.2) | 54.2 (0.6) | 32.8 (0.4) | 31.0 (0.8) |

Table 15: Performance on additional WILDS benchmark datasets. The DL column indicates whether the algorithm uses domain labels. Using a Shared-Base ensemble consistently results in gains in OOD generalization metrics over prior methods. However, we observe no further benefits from reweighting the ensemble via HYRE on these datasets.

All architectures are trained end-to-end by minimizing the sum of a standard loss function (cross-entropy for classification, MSE for regression) over all ensemble members:

$$\sum_{h=1}^{H} \mathcal{L}\left(f_h(x), y\right). \tag{4}$$

Here, $x$ is an input example, $y$ is the true label, and $f^i$ is the $i$-th ensemble member. While each individual model minimizes the training loss, we want the ensemble members to extrapolate to unseen data in diverse ways. The specific ensemble parameterizations, which we describe below, are designed to achieve this goal.

### E.1 Vanilla Ensemble

A vanilla ensemble consists of $H$ independently initialized and trained neural networks with identical architectures. Each network $f_h$ takes an input $x$ and produces an output $f_h(x)$. No parameters are shared. While simple to implement, this approach scales poorly as $H$ increases since both memory and computation scale linearly with $H$.

### E.2 Shared-Base Ensemble

We propose a scalable neural network architecture that can represent thousands of diverse ensemble members. The network outputs $H$ real-valued predictions in parallel, with the output space being $\mathbb{R}^H$. The architecture comprises a frozen prior network $f_p$ and a learnable network $f_\theta$, both of which produce outputs of shape $\mathbb{R}^H$. Although the architectures of $f_p$ and $f_\theta$ are identical in our experiments, this is not a requirement.

For a given input $x$, the network output is

$$f^p(z) + f^\theta(z) = \begin{bmatrix} f_1^p(z) + f_1^\theta(z) \\ f_2^p(z) + f_2^\theta(z) \\ \vdots \\ f_H^p(z) + f_H^\theta(z) \end{bmatrix} \in \mathbb{R}^H \tag{5}$$

where each prediction $f_i^p(z) + f_i^\theta(z)$ is compared against the ground-truth label $y$. The parameters of $f^p$ are fixed at initialization and do not change during training; the parameters of $f^\theta$ are learnable.

Using the frozen prior network $f^p$ is crucial to the diversity in this architecture. If we were to only train $f^\theta$, the ensemble of the $H$ predictions would have low diversity due to co-adaptation. To understand why this

architecture produces a diverse ensemble, note that each learnable head solves a shifted task determined by the corresponding prior network head. Since we undo this shifting when producing the final prediction, we can view the different learnable heads as solving a different yet equivalent task.

### E.3  Epinet

The epinet architecture combines a base model $f^{\text{base}} : \mathcal{X} \to \mathbb{R}^K$ with an epistemic network $f^{\text{epi}} : \mathcal{Z} \times \mathbb{R}^{d_{\text{ftrs}}} \times \mathcal{X} \to \mathbb{R}^K$. The base model can be any regular neural network, including a large pretrained model, and is used to extract features through a feature extractor $\phi : \mathcal{X} \to \mathbb{R}^{d_{\text{ftrs}}}$. Here, $d_{\text{ftrs}}$ is the dimension of the extracted intermediate representations.

The epistemic network (epinet) is composed of two parts:

- A frozen prior network $f^{\text{epi-frozen}} : \mathcal{X} \to \mathbb{R}^{1,\dots,d_{\text{index}} \times K}$. The parameters of this network are fixed at initialization and do not change during training.

- A trainable network $f^{\text{epi-trainable}} : \mathcal{Z} \times \mathbb{R}^{d_{\text{ftrs}}} \times \mathcal{X} \to \mathbb{R}^K$.

Given an epistemic index $z \in \mathbb{R}^d$ and input $x \in \mathcal{X}$, we compute the model output as:

$$f(z,x) = f^{\text{base}}(x) + v f^{\text{epi-frozen}}(x) \cdot z + f^{\text{epi-trainable}}(z, \phi(x), x) \cdot z \tag{6}$$

where $\cdot$ is the dot product and $v \in (0, \infty)$ is the so-called prior scale. At each step, we sample multiple epistemic indices $z$ to form an ensemble, i.e., $f_1(x), \dots, f_H(x) = f(z_1, x), \dots, f(z_H, x)$. This architecture efficiently generates diverse predictions by sampling different epistemic indices $z$ while leveraging a potentially large pretrained base model.

## F  Repulsion vs Random Priors for Diversity

A line of prior work uses repulsion to enforce diversity among ensemble members by adding a regularization term that favors sufficiently "different" predictions according to some distance metric. For example, Teney et al. [69] uses a repulsion term that maximizes the cosine distance between the gradient of each ensemble member, and Lee et al. [43] maximizes the mutual information of ensemble predictions on OOD inputs. While these techniques have succeeded in some settings, our early experiments suggest that such explicit regularization often yields a suboptimal ensemble. The repulsion term can overpower the learning signal in the training data, leading to ensemble members that are diverse but inaccurate.

In contrast, diversification via random priors [50] provides a more balanced approach. The key idea is to initialize each ensemble member with a different random prior function that remains fixed throughout training. This introduces diversity from the start without explicitly optimizing for it during training. This approach maintains diversity without sacrificing training accuracy, and the degree of diversification is easily controlled by scaling the prior functions.

## G  Function-Space Dimensionality Reduction

Here, we expand on the idea of PCA on ensemble predictions. A central challenge with large model ensembles is understanding the commonalities and differences among the individual models. The high-level idea is that PCA applied to ensemble predictions reveals the major direction of variation within an ensemble of models. This dimensionality reduction allows us to clearly interpret model behaviors and identify groups of related datapoints Additionally, PCA enables the generation of new functions with similar statistical properties by parameterizing a low-rank Gaussian distribution in the joint prediction space, which we can sample from.

### G.1 Motivating Example

Consider three models $f_1, \ldots, f_3$ and five inputs $z_1, \ldots, z_5$. Denoting each model's predicted probability for an input as $p_{nh} = \sigma(f_h(z_n)) \in [0,1]$, assume that the matrix of predictions is

$$
\begin{pmatrix} p_{11} & p_{12} & p_{13} & p_{14} & p_{15} \\ p_{21} & p_{22} & p_{23} & p_{24} & p_{25} \\ p_{31} & p_{32} & p_{33} & p_{34} & p_{35} \end{pmatrix} = \begin{pmatrix} 1 & 0 & 1 & 0 & 1/2 \\ 0 & 1 & 1/2 & 1/2 & 1/2 \\ 1/2 & 1/2 & 0 & 1 & 1/2 \end{pmatrix}. \tag{7}
$$

Each row of this matrix shows one model's prediction on the entire pool of inputs, and each column shows every model's prediction on a single input. We can analyze such a matrix of predictions on three levels, each revealing increasing amounts of structure within the ensemble:

**Level 1: Per-sample ensemble uncertainty.** We can first compute the average prediction $\bar{p}(x) = \frac{1}{H} \sum_h p_{nh}$ for each datapoint. For the predictions in (7), the average prediction is $\bar{p}(x) = 1/2$ for every input $x$, and thus the collection of models may be viewed as equally uncertain about each of the 5 inputs. This is the measure of ensemble uncertainty commonly used for ensembles [40].

**Level 2: Per-sample disagreement.** We can further account for the amount of disagreement among ensemble members for each datapoint. Note that for the four inputs $z_1, z_2, z_3, z_4$, there is strong disagreement between two functions where one predicts 0 and the other predicts 1. This is not true of $z_5$, where all functions predict $1/2$. Uncertainty metrics that take disagreement into account, such as the BALD criterion [27], will reveal that the ensemble is more uncertain about $z_1, z_2, z_3, z_4$ than it is about $z_5$.

**Level 3: Joint predictions.** First, note that the two approaches above discard all information about which ensemble member made which individual prediction for a given input, by (1) averaging all predictions or (2) considering only the unordered set of predictions. There is additional structure to the differences among ensemble members that we can extract by considering the joint predictions, i.e., viewing each column of (7) as an object in itself. The pair of inputs $(z_1, z_2)$ are closely related since they deviate from the ensemble prediction in the same "direction" in the joint prediction space ($\mathbb{R}^H$). We can make the same observation about the pair $(z_3, z_4)$. To see this structure more clearly, consider the matrix of deviations from the ensemble prediction $\delta_{nh} = p_{nh} - \frac{1}{H} \sum_h p_{nh}$:

$$
\begin{pmatrix} \delta_{11} & \delta_{12} & \delta_{13} & \delta_{14} & \delta_{15} \\ \delta_{21} & \delta_{22} & \delta_{23} & \delta_{24} & \delta_{25} \\ \delta_{31} & \delta_{32} & \delta_{33} & \delta_{34} & \delta_{35} \end{pmatrix} = \frac{1}{2} \begin{pmatrix} 1 & -1 & 1 & -1 & 0 \\ -1 & 1 & 0 & 0 & 0 \\ 0 & 0 & -1 & 1 & 0 \end{pmatrix}. \tag{8}
$$

This clearly shows that the vector of joint deviations $(\delta_{11}, \delta_{12}, \delta_{13})$ is the negative of that of $(\delta_{21}, \delta_{22}, \delta_{23})$. More generally, we can view the vector of deviations $(\delta_{1n}, \delta_{2n}, \delta_{3n})$ as a representation of the datapoint $z_n$ in the joint prediction space. In this sense, the matrix of predictions $\{p_{nh}\}$ can be explained by the mean prediction 0.5 for each datapoint, together with two factors of variation $(1, -1, 0)$ and $(1, 0, -1)$ appropriately applied to each input. We next describe how to automatically extract such consistent high-level factors in an ensemble from the matrix of predictions.

### G.2 PCA on Ensemble Predictions

We propose to apply PCA to the $H \times N$ matrix of residual predictions to obtain $P$ principal components. Each principle component is a vector of size $H$ that captures the orthogonal factors of variation in how ensemble members extrapolated from the training data. Given a set of weights $w_1, \ldots, w_P$ over principal components, we can "reconstruct" a set of joint predictions as

$$
p(x) = \bar{p}(x) + \begin{pmatrix} w_1 & \cdots & w_P \end{pmatrix} \begin{pmatrix} c_{11} & \cdots & c_{1H} \\ c_{21} & \cdots & c_{2H} \\ \vdots & \ddots & \vdots \\ c_{P1} & \cdots & c_{PH} \end{pmatrix} \begin{pmatrix} p_1(x) - \bar{p}(x) \\ p_2(x) - \bar{p}(x) \\ \vdots \\ p_H(x) - \bar{p}(x) \end{pmatrix}, \tag{9}
$$

where we denote the mean prediction as $\bar{p}(x) = \frac{1}{H} \sum_h p_{nh}$ and the $P$ principal components as $C \in \mathbb{R}^{P \times H}$.

We highlight two known interpretations of PCA that have interesting implications for our goal of summarizing ensemble predictions:

**Maximum mutual information / variance after projection.** PCA finds the linear projection $y = w^\top x$ with unit vector $w$ that achieves maximum mutual information $I(x; y)$, or equivalently, maximum variance $\text{Var}(y)$. Each principal component finds the linear combination of ensemble members that preserves the most information about the set of joint ensemble predictions. This is closely related to the disagreement term in Bayesian active learning [27].

**Factor model.** The principal components are maximum likelihood parameters under a linear Gaussian factor model of the data [70]. Indeed, we can view our principal components as orthogonal modifications to the mean prediction $\bar{p}(x)$. The distribution of ensemble members is closely approximated by "reconstructed predictions" (9), where $z_{1:P} \sim \mathcal{N}(0, \mathrm{I}^P)$. We can view each principal component as a consistent high-level direction of functional variation in which the training data provided insufficient information.

## H  PERSONA Dataset Details

Below, we list the personas used in our PERSONA [9] experiments. The dataset includes 1000 personas in total, each with 200 preference pairs. We subsampled 10 personas from the original dataset of 1000, ensuring a diverse set of backgrounds, ages, and lifestyles.

**Persona 1.** Age: 1. Sex: Male. Race: White alone. Ancestry: Irish. Household language: English only. Education: Not applicable. Employment status: Not applicable. Class of worker: Not applicable. Industry category: Not applicable. Occupation category: Not applicable. Detailed job description: Not applicable. Income: Not applicable. Marital status: Too young to be married. Household type: Cohabiting couple household with children of the householder less than 18. Family presence and age: With related children under 5 years only. Place of birth: Missouri/MO. Citizenship: Born in the United States. Veteran status: Not applicable. Disability: None. Health insurance: With health insurance coverage. Fertility: Not applicable. Hearing difficulty: None. Vision difficulty: None. Cognitive difficulty: None. Ability to speak english: Not applicable. Big five scores: Openness: High, Conscientiousness: High, Extraversion: Low, Agreeableness: Extremely High, Neuroticism: Extremely Low. Defining quirks: Loves to play with his food. Mannerisms: Waves hands when excited. Personal time: Spends most of his time playing, sleeping, and learning to walk. Lifestyle: Lives a carefree and playful lifestyle. Ideology: Not applicable. Political views: Not applicable. Religion: Other Christian.

**Persona 2.** Age: 11. Sex: Male. Race: White alone. Ancestry: Irish. Household language: English only. Education: Grade 4. Employment status: Unemployed. Class of worker: Not applicable. Industry category: Not applicable. occupation category: Not applicable Detailed job description: Student. Income: 0. Marital status: Never married or under 15 years old. Household type: Cohabiting couple household with children of the householder less than 18. Family presence and age: With related children 5 to 17 years only. Place of birth: Louisiana/LA. Citizenship: Born in the United States. Veteran status: Not applicable. Disability: None. Health insurance: With health insurance coverage. Big five scores: Openness: Low, Conscientiousness: Low, Extraversion: High, Agreeableness: High, Neuroticism: Average. defining quirks: Loves to draw and create stories Mannerisms: Often seen doodling or daydreaming. Personal time: Spends free time drawing or playing video games. Lifestyle: Active and playful, enjoys school and spending time with friends. Ideology: Undeveloped. Political views: Undeveloped. Religion: Religiously Unaffiliated.

**Persona 3.** Age: 19. Sex: Male. Race: Asian Indian alone. Ancestry: Indian. Household language: Hindi. Education: 1 or more years of college credit, no degree. Employment status: Not in labor force. Class of worker: Not Applicable. Industry category: Not Applicable. Occupation category: Not Applicable. Detailed job description: Not Applicable. Income: -60000.0. Marital status: Never married or under 15 years old. Household type: Living with parents. Family presence and age: Living with two parents. Place of birth: India. Citizenship: Not a U.S. citizen. Veteran status: Non-Veteran. Disability: None. Health insurance: With health insurance coverage. Big five scores: Openness: Average, Conscientiousness: High, Extraversion: Extremely Low, Agreeableness: Extremely High, Neuroticism: Extremely Low. defining quirks: Passionate about music Mannerisms: Expressive hand gestures when speaking. Personal time: Practicing music or

studying. Lifestyle: Student and Music Enthusiast. Ideology: Liberal. Political views: Liberal. Religion: Other Christian.

**Persona 4.** Age: 29. Sex: Female. Race: Laotian alone. Ancestry: Laotian. Household language: Asian and Pacific Island languages. Education: Some college, but less than 1 year. Employment status: Armed forces, at work. Class of worker: Federal government employee. Industry category: MIL-U.S. Navy. Occupation category: MIL-Military Enlisted Tactical Operations And Air/Weapons Specialists And Crew Members. Detailed job description: Maintains and operates tactical weapons systems. Income: 81000.0. Marital status: Married. Household type: Married couple household with children of the householder less than 18. Family presence and age: With related children 5 to 17 years only. Place of birth: California/CA. Citizenship: Born in the United States. Veteran status: Now on active duty. Disability: None. Health insurance: With health insurance coverage. Big five scores: Openness: Average, Conscientiousness: High, Extraversion: Average, Agreeableness: High, Neuroticism: Average. Defining quirks: Collects military memorabilia. Mannerisms: Frequently uses military jargon. Personal time: Spends time with family and collecting military memorabilia. Lifestyle: Disciplined and active. Ideology: Conservative. Political views: Republican. Religion: Protestant.

**Persona 5.** Age: 36. Sex: Female. Race: Some Other Race alone. Ancestry: Hispanic. Household language: English. Education: Regular high school diploma. Employment status: Civilian employed, at work. Class of worker: Employee of a private for-profit company or business, or of an individual, for wages, salary, or commissions. Industry category: FIN-Insurance Carriers. Occupation category: OFF-Insurance Claims And Policy Processing Clerks. Detailed job description: Processes insurance claims and policies. Income: 182000.0. Marital status: Married. Household type: Married couple household with children of the householder less than 18. Family presence and age: With related children under 5 years only. Place of birth: New Mexico/NM. Citizenship: Born in the United States. veteran status: Non-Veteran Disability: None. Health insurance: With health insurance coverage. Big five scores: Openness: Extremely Low, Conscientiousness: Extremely High, Extraversion: Extremely High, Agreeableness: High, Neuroticism: Average. Defining quirks: Enjoys bird-watching. Mannerisms: Often taps foot when thinking. Personal time: Spends free time with family or in nature. Lifestyle: Active and family-oriented. Ideology: Conservative. Political views: Republican. Religion: Other Christian.

**Persona 6.** Age: 44. Sex: Female. Race: Black or African American alone. Ancestry: Haitian. household language: Other Indo-European languages education: Associate's degree Employment status: Civilian employed, at work. Class of worker: Employee of a private not-for-profit, tax-exempt, or charitable organization. Industry category: FIN-Banking And Related Activities. Occupation category: OFF-Tellers. Detailed job description: Handles customer transactions at the bank, including deposits, withdrawals, and loan payments. Income: 40000.0. Marital status: Separated. Household type: Female householder, no spouse/partner present, with children of the householder less than 18. Family presence and age: With related children 5 to 17 years only. Place of birth: Haiti. Citizenship: Not a U.S. citizen. Veteran status: Non-Veteran. Disability: None. Health insurance: With health insurance coverage. Big five scores: Openness: High, Conscientiousness: Extremely Low, Extraversion: Average, Agreeableness: Average, Neuroticism: Extremely Low. Defining quirks: Loves to cook Haitian cuisine. Mannerisms: Often taps her foot when stressed. Personal time: Taking care of her children, Pursuing further education. Lifestyle: Busy, Family-oriented. Ideology: Egalitarian. Political views: Democrat. Religion: Protestant.

**Persona 7.** Age: 52. Sex: Female. Race: Korean alone. Ancestry: Korean. Household language: Asian and Pacific Island languages. Education: Regular high school diploma. Employment status: Civilian employed, at work. Class of worker: State government employee. Industry category: ENT-Restaurants And Other Food Services. Occupation category: EAT-First-Line Supervisors Of Food Preparation And Serving Workers. Detailed job description: Supervises food preparation and serving workers in a state government facility. Income: 133900.0. Marital status: Married. Household type: Married couple household, no children of the householder less than 18. Family presence and age: No related children. Place of birth: Korea. Citizenship: U.S. citizen by naturalization. Veteran status: Non-Veteran. Disability: None. Health insurance: With health insurance coverage. big five scores: Openness: Average, Conscientiousness: Extremely High, Extraversion: Extremely Low, Agreeableness: Extremely Low, Neuroticism: Average defining quirks: Deep love for literature and reading Mannerisms: Constantly adjusts her glasses. Personal time: Spends free time

reading or engaging in community activism. Lifestyle: Quiet and community-oriented. Ideology: Liberal. Political views: Democratic. Religion: Protestant.

**Persona 8.** Age: 58. Sex: Male. Race: White. Ancestry: Scottish. Household language: English. Education: Bachelor's Degree. Employment status: Employed. Class of worker: Private. industry category: Investigation And Security Services Occupation category: Sales Manager. Detailed job description: Oversees sales teams, sets sales goals, and develops strategies to achieve these goals. Income: 198200. Marital status: Married. Household type: Married couple household, no children under 18. Family presence and age: No related children. Place of birth: Florida. Citizenship: US Citizen. veteran status: Non-Veteran Disability: With a disability. Health insurance: With health insurance coverage. Big five scores: Openness: High, Conscientiousness: Extremely High, Extraversion: Average, Agreeableness: Average, Neuroticism: Average. Defining quirks: Keen interest in security technology and crime novels. mannerisms: Constantly checks his surroundings Personal time: Researching the latest security technologies or enjoying a round of golf. Lifestyle: Active and health-conscious. Ideology: Conservative. Political views: Republican. Religion: Catholic.

**Persona 9.** Age: 65. Sex: Female. Race: White alone. Ancestry: Italian. Household language: Other Indo-European languages. Education: Master's degree. Employment status: Civilian employed, at work. Class of worker: Self-employed in own incorporated business, professional practice or farm. Industry category: ENT-Traveler Accommodation. Occupation category: FIN-Accountants And Auditors. Detailed job description: Manages financial records and tax data for her own travel accommodation business. Income: 188600.0. Marital status: Married. Household type: Married couple household, no children of the householder less than 18. Family presence and age: No related children. Place of birth: Delaware/DE. Citizenship: Born in the United States. Veteran status: Non-veteran. Disability: None. Health insurance: With health insurance coverage. ability to speak english: Well. Big five scores: Openness: Average, Conscientiousness: Low, Extraversion: Low, Agreeableness: Average, Neuroticism: Extremely High. Defining quirks: Has an extensive collection of vintage travel posters. Mannerisms: Tends to use Italian phrases in conversation. Personal time: Spends her free time exploring new places, trying new cuisines, and learning about different cultures. Lifestyle: Leads a busy lifestyle managing her business, but always finds time for her passion for travel and culture. Ideology: Believes in the importance of understanding and appreciating different cultures. Political views: Liberal. Religion: Protestant.

**Persona 10.** Age: 75. Sex: Female. Race: White alone. ancestry: Scottish Household language: English only. Education: Professional degree beyond a bachelor's degree. Employment status: Not in labor force. Class of worker: Retired. Industry category: Healthcare. Occupation category: Doctor. Detailed job description: Retired pediatrician. Income: 98000.0. Marital status: Never married. Household type: Female householder, no spouse/partner present, living alone. Family presence and age: No family. Place of birth: Massachusetts/MA. citizenship: Born in the United States veteran status: Non-Veteran Disability: None. Health insurance: With health insurance coverage. Big five scores: Openness: Average, Conscientiousness: Average, Extraversion: High, Agreeableness: Extremely High, Neuroticism: Average. Defining quirks: Enjoys cooking traditional Scottish meals. Mannerisms: Often hums traditional Scottish tunes. Personal time: Spends free time volunteering at the local church and community center. Lifestyle: Active but relaxed, with a focus on maintaining health and staying involved in the community. Ideology: Conservative. Political views: Republican. Religion: Catholic.

