# OpenReview forum: "Inference-Time Alignment via Hypothesis Reweighting"
_TMLR — Accepted by TMLR_

### Review · Reviewer_Gs1K · 2026-01-06

**Summary Of Contributions:**

The paper introduces Hypothesis Reweighting (HyRe), a simple and efficient method for inference-time alignment and personalization of machine learning models under distribution shift and preference ambiguity. The core idea is to train a lightweight ensemble of prediction heads on a shared backbone, where each head represents a different hypothesis consistent with the training data, and then reweight these hypotheses at inference time using only a small number (1–5) of labeled examples from the target distribution.
The reweighting is performed via a generalized Bayesian update over ensemble members based on task-relevant performance metrics (e.g., 0–1 accuracy), requiring only a single forward pass and incurring <1% computational overhead. This makes HyRe practical for real-time personalization scenarios where fine-tuning or large-context prompting is infeasible.
Empirically, the paper demonstrates that:
- Uniform ensemble averaging often underperforms the best-aligned individual hypothesis under distribution shift.
- HyRe significantly improves performance in low-data regimes across regression and preference learning tasks.
- On large-scale benchmarks such as RewardBench, HyRe achieves state-of-the-art results at both 2B and 8B scales with only a handful of adaptation examples, outperforming much larger reward models.
- Improvements in reward model accuracy translate to measurable gains in downstream generation quality.

Overall, the work positions inference-time ensemble reweighting as a principled, scalable alternative to fine-tuning and few-shot prompting for alignment and personalization.

**Strengths:**
- The method is easy to implement (reweighting ensemble heads via a softmax update) and delivers gains across diverse tasks.
- Requires only 1–5 labeled examples and a single forward pass, making it viable for real-time, per-user personalization.
- Demonstrates state-of-the-art performance on RewardBench at 2B and 8B scales, outperforming larger and more expensive models.
- Casting reweighting as generalized Bayesian inference provides theoretical clarity and robustness, especially with non-differentiable metrics like classification accuracy.
- Performs robustly even when adaptation data is noisy, skewed, or drawn from multiple preference distributions.
Ensemble heads add a negligible number of parameters (≈0.03% for a 2B model), with no changes to the backbone at inference time.

**Weaknesses:**
- The approach assumes that the ensemble already spans the relevant hypotheses for the target distribution; if the correct behavior lies outside this span, reweighting cannot recover it.
- Results suggest that improvements are strongest for personalization and preference ambiguity, while benefits are more modest for large, structural distribution shifts (e.g., some WILDS settings).
- Although minimal, HyRe still needs labeled examples at test time, which may not always be available or cheap to obtain.
- The method operates on a small batch of adaptation data and does not directly address continual or single-sample online (streaming) updates.
- Ensemble diversity is not explicitly enforced during training; while it works empirically, theoretical guarantees or stronger diversity mechanisms could improve robustness.
- The paper lacks a discussion of theoretical guarantees or related theoretical work.

**Audience:**

Yes

**Audience Explanation:**

Yes, at least some individuals in TMLR’s audience would be interested in the findings, particularly those working on alignment and preference learning, inference-time or test-time adaptation, ensemble-based uncertainty or model selection. However, the appeal is not broad across the entire TMLR readership. The paper is likely to resonate most strongly with a subset of applied and alignment-focused researchers, rather than with readers seeking new theoretical frameworks or general-purpose learning principles.

**Broader Impact Concerns:**

While the work is primarily methodological, its potential applications in personalization and alignment at scale raise some ethical considerations. The paper would benefit from a clearer Broader Impact Statement discussing safeguards, limitations, and responsible deployment practices, particularly around safety degradation (e.g. robustness to strategic preferences), misuse (potential for unequal or harmful use), and governance of inference-time adaptation.

**Claims And Evidence:**

No

**Claims Explanation:**

- The method reweights a trained ensemble using a Bayesian-like update with 1–5 labeled points. The paper shows empirical improvements on several tasks and benchmarks, including preference and regression settings, and strong performance on RewardBench. While results are promising, much of the evaluation focuses on few tasks and a specific benchmark suite. The evidence does not yet convincingly demonstrate that the reweighting approach generalizes to widely differing task structures or complex real-world distributions beyond these benchmarks. Additionally, distribution shift cases could be more thoroughly stress-tested with stronger baselines.

- The paper includes experiments on personalization and some distribution shift settings. But the experiments do not convincingly cover the domain of underspecification as broadly as suggested.
- The main comparisons are often to uniform ensembles or larger frozen models. There is limited comparison with adaptive inference methods or meta-learning approaches, which can also personalize with few examples. Without these, it is hard to discern whether the observed improvements are specific to HyRe or common to any adaptation technique in these regimes.
- The reported gains are clear in selected metrics, but the paper often lacks confidence intervals, statistical significance tests, or variance analyses that would make empirical claims more convincing.
- It is unclear if all code, training details, and adaptation protocols will be readily available or easily reproducible.

**Requested Changes:**

The following changes would help strengthen the work in my view:

- The paper currently makes broad claims about inference-time alignment and distribution shift, but the strongest evidence is limited to personalization and preference-based settings. The authors should narrow or explicitly qualify claims to clarify when HyRe is expected to work (e.g., preference ambiguity, underspecification) and when it is not (e.g., severe covariate or semantic shift). A dedicated discussion of failure cases where reweighting does not help (or hurts) is essential to align claims with evidence.
- Comparisons to other test-time adaptation methods, lightweight fine-tuning variants, or learned ensemble weighting/gating approaches are recommended. Without these, it is unclear whether the gains are specific to HyRe or reflect more general adaptation effects.
- Many experimental results are presented without measures of variability. The authors should report confidence intervals, standard deviations, or statistical significance tests throughout.
- HyRe fundamentally relies on the ensemble spanning relevant hypotheses, yet diversity is not explicitly enforced or analyzed. The paper should clearly articulate this dependency and discuss scenarios where insufficient diversity would cause the method to fail. Ideally, this should be supported by ablation studies varying ensemble size or diversity strength.
- The generalized Bayesian reweighting is elegant but under-analyzed. A sensitivity study on loss choice, temperature/scale parameters, and adaptation set size would improve clarity and reproducibility.
- A deeper comparison with MoE would help strengthen the positioning of the paper.

Minor:
- Second para of introduction: "inference-time methods that can adapt without gradient updates or *scale with* large context windows."

---

> ### Author Response · Authors · 2026-02-16
> **Response to Reviewer Gs1K**
>
> We thank the reviewer for the thorough review.
>
> > "The paper currently makes broad claims about inference-time alignment and distribution shift..."
>
> We narrowed the Introduction and Abstract to focus on personalization and preference ambiguity. The Discussion now explicitly discusses our negative results on the WILDS benchmark: across four datasets, reweighting did not outperform the uniform ensemble. We hypothesize these shifts are structural rather than preference-based, scoping HyRe's applicability to personalization settings.
>
> > "HyRe fundamentally relies on the ensemble spanning relevant hypotheses, yet diversity is not explicitly enforced or analyzed."
>
> We have added a new experiment that explicitly measures ensemble diversity (Table 13 in the appendix). Here, we report $K_\text{eff}$ (perplexity of the weight distribution). On average, 43 of 100 heads carry meaningful weight after adaptation.
>
> We agree this deserves explicit discussion and have updated Section 7. We do not have general guarantees against collapse; we enforce diversity architecturally (prior networks; Osband et al., 2018). This is an empirically effective approach for maintaining diversity in ensembles. Our paper provides evidence that prior networks indeed meaningfully increase diversity in the settings we consider: prior scale controls per-head sharpness (Figure 4), and gains across 32 datasets (Figure 5) would be impossible under collapse.
>
> > "There is limited comparison with adaptive inference methods or meta-learning approaches..."
>
> The paper includes controlled comparisons at the same N: few-shot prompting (Table 4), fine-tuning (Table 2), and four alternative ensemble weighting methods (Table 8). HyRe outperforms all, so the gains are specific to the Bayesian reweighting scheme we use.
>
> Unlike MAML or prototypical networks, HyRe requires no episodic meta-training; diversity comes from prior networks during standard training. We discuss these connections in the related work section.
>
> > "A sensitivity study on loss choice, temperature/scale parameters, and adaptation set size..."
>
> We would like to first point out that sensitivity studies on loss choice (Table 5, accuracy vs. cross-entropy) and adaptation set size (Table 3, Figure 6...) were already in the original submission.
>
> We added a temperature sweep on RewardBench. The default $\tau{=}1$ performs best at $N \geq 5$; low temperatures collapse to hard selection, whereas high temperatures flatten toward uniform.
>
> | $\tau$ | $N{=}1$ | $N{=}2$ | $N{=}5$ | $N{=}10$ | $N{=}20$ |
> |--------|---------|---------|---------|----------|----------|
> | 0.01 | **92.88** | 92.74 | 93.06 | 92.87 | 92.90 |
> | 0.10 | 92.85 | 92.69 | 92.97 | 92.96 | 93.09 |
> | 0.50 | 92.79 | 92.76 | **93.12** | 93.42 | 93.54 |
> | **1.00** | 92.73 | **92.90** | **93.12** | **93.44** | **93.72** |
> | 2.00 | 92.56 | 92.73 | 92.97 | 93.18 | 93.50 |
> | 5.00 | 92.58 | 92.68 | 92.85 | 92.99 | 93.24 |
> | 10.0 | 92.40 | 92.47 | 92.56 | 92.70 | 92.85 |
>
> > "A deeper comparison with MoE..."
>
> MoE uses per-token gating to improve compute efficiency; a single expert is not intended to be a standalone model. HyRe applies a global weighting across complete, standalone reward models to resolve preference ambiguity. We clarify this distinction in the revision.
>
> > "The paper would benefit from a clearer Broader Impact Statement..."
>
> Thank you for raising this. We now address adversarial adaptation data risks, noting that explicit weights enable auditing and describing concrete mitigations (filtering adaptation data and constraining permissible feedback types).
>
> > "It is unclear if all code, training details, and adaptation protocols will be readily available..."
>
> Appendix C provides comprehensive details on our experimental setup (model URLs, hyperparameters, ensemble configuration). We added a reproducibility statement and will release code upon acceptance.

---

### Review · Reviewer_ckCB · 2026-01-11

**Summary Of Contributions:**

In this paper, the authors study the problem of preference modeling / personalization, where existing methods like SFT are too slow, require extensive data and are computationally expensive. The authors propose Hypothesis Reweighting (HyRe), which a lightweight method to align models to a specific user or domain at inference time.In HyRE, the authors train multiple prediction heads upon a single backbone model, and each head learns different interpretation of the training data. At test time, HyRE will use a few examples to evaluate how well each head predicts them and doing a weighted average to align with the examples. As such, HyRe requires only some examples and one single forward pass to perform inference. The authors evaluated on different becnhmarks, with a focus on reward modeling, where the proposed method can outperform existing baselines and show improved generation with methods like BoN sampling.

Pros:
1. HyRe is an inference-time method that trains only a few heads with negligible overhead. It only requires similar test-time computation, and thus avoiding extensive costs of training or multi-stage prompting.
2. Ther performance of HyRe is impressive on reward modeling benchmarks, with only a few examples, the proposed method can outperform larger baselines and achieve substantial gains compared to the base model.

Cons:
1. The proposed method does not appear substantially novel for test-time scaling or reward modeling, imo, the authors offer an incremental improvement rather than introducing a new test-time paradigm.
2. In the main experiments, the comparison is not exactly fair since the proposed method trains the heads on UltraFeedback and in addition, inference will require additional few-shot examples where baselines do not.
3. When using the uniform option in HyRe, there's basically not gains on the benchmarks, this suggest that the main performance improvements are from the few-shot examples rather than the proposed methodology. Similarly in Table 8, the proposed method does not show substantial gains when the number of examples are limited.

**Audience:**

Yes

**Audience Explanation:**

Yes, the findings are relevant for research communities in reward modeling / personalization etc.

**Claims And Evidence:**

Yes

**Claims Explanation:**

Yes, the proposed method and authors's claims are accurate.

**Requested Changes:**

1. The authors should benchmark against more & recent generative models  (i.e., similar to Table 4) , such as Llama 3.1 70B and Qwen 3 4/8B, to ensure a comprehensive evaluation on HyRe and few-shot generative reward modeling.

2. I'd appreciate if the authors can extend their discussions on where exactly the gains are from, are these from the trained heads or rather from the newly introduced demonstrations during inference?

---

> ### Author Response · Authors · 2026-02-16
> **Response to Reviewer ckCB**
>
> We thank the reviewer for the clear and constructive feedback.
>
> > "I'd appreciate if the authors can extend their discussions on where exactly the gains are from..."
> >
> > "When using the uniform option in HyRe, there's basically not gains on the benchmarks..."
>
> We added a "Source of improvements" paragraph (Section 6.4) that organizes our evidence on the sources of the gains in HyRe.
> 1. The uniform ensemble does not improve over the base model (Table 3);
> 2. HyRe outperforms few-shot prompting at the same N (Table 4);
> 3. HyRe outperforms fine-tuning at low N (Table 2, Figure 4);
> 4. HyRe outperforms all 4 alternative weighting methods at the same N (Table 8).
>
> Specifically for Table 8, even at N=5, HyRe (85.73%) outperforms GEM (84.49%) at N=40. At the same number of adaptation samples, HyRe greatly outperforms it (87.74%)
>
> > "In the main experiments, the comparison is not exactly fair since the proposed method trains the heads on UltraFeedback..."
>
> HyRe's heads are trained on UltraFeedback, a standard preference dataset also used by the baseline reward models. All controlled comparisons (Tables 2, 4, 8) use the same N adaptation examples. Table 3 compares against the RewardBench leaderboard scores (N=0 by convention) because no established few-shot reward-model adaptation methods exist; this is precisely the gap HyRe addresses.
>
> > "The authors should benchmark against more & recent generative models..."
>
> Thank you for the suggestion. Table 3 already includes Llama 3.1 70B (INF-ORM, 95.1%), Claude-3 Sonnet (84.2%), GPT-4 (86.7%), and Gemini-1.5-Pro (86.8%). We add the strongest Qwen-family generative model on the RewardBench leaderboard, RISE-Judge-Qwen2.5-7B (88.2%).
>
> > "The proposed method does not appear substantially novel... the authors offer an incremental improvement."
>
> We respectfully disagree with the characterization. We demonstrate that a closed-form reweighting with 5 examples and <1% overhead can improve alignment at inference time, even beating much larger models (GPT-4 at 86.7% vs HyRe-8B at 95.0%). To our knowledge, no prior work demonstrates real-time per-user reward model adaptation at this scale without fine-tuning. We validate this across 32 datasets at 2B and 8B scale.

---

### Review · Reviewer_z4ps · 2026-02-02

**Summary Of Contributions:**

### The contributions of the paper
This paper proposes HyRe, a method that addresses the limitations of fine-tuning and in-context learning in scenarios requiring sub-second adaptation for concurrent users by employing an ensemble approach. The authors aim to overcome the limitations of conventional uniform ensemble averaging through generalized Bayesian inference. The method enables real-time personalization of reward models by training multiple prediction heads on a shared backbone and performing Bayesian reweighting at inference time using labeled examples drawn from the target distribution, achieving improved performance on RewardBench.

---

### Strong points

1. The authors appropriately present the limitations of existing fine-tuning and in-context learning approaches, along with observations about ensemble methods.  In their introduction, they provide a motivated background for their proposed method.

2. The method requires only a single forward pass for adaptation, and makes it practically applicable in real-world scenarios where fine-tuning is prohibitive.

3. HyRe consistently outperforms state-of-the-art reward models on RewardBench at both the 2B and 8B parameter scales.

4. The authors describe the differentiation from MoE, pluralistic alignment methods, and inference-time alignment approaches, providing a clear positioning of the paper within the existing literature.

5. The authors acknowledge the limitations of their work from three perspectives: scope of applicability, labeling requirements, and ensemble coverage.

---

### Weak points


1. **Ensemble diversity is assumed rather than guaranteed.** The authors use random initialization to ensure diversity, but they do not provide theoretical guarantees or systematic analysis of when sufficient diversity will emerge. (While Figure 5 shows that the diversity coefficient matters, this analysis remains incomplete.)

2. **The motivation requires refinement.** The authors present "sub-second adaptation for millions of concurrent users" as the motivation in the introduction, but whether the problem they are actually solving truly addresses this scenario remains ambiguous. This presentation creates a gap between "what to solve" and "how to solve."

3. **Insufficient analysis of ensemble collapse.** The authors discuss ensemble collapse but empirically demonstrate only that diversity is achieved, providing no explanation or analysis of when and under what conditions it is ensured.

4. **The paper requires improvement in writing quality.** In particular, the notation for algorithms and equations is inconsistent, making the paper difficult to read. Additionally, many contents lack proper references to the Appendix, making it difficult to understand why certain materials are placed in the Appendix. Important information that should be in the main text exists only in the Appendix, or is missing from the Appendix entirely.

**Additional Comments:**

1. In Figure 1, since the authors use $H=100$ heads, it would be better to generalize with $H$ heads or explicitly indicate 100 heads rather than depicting only 4 heads. Additionally, the frozen prior network, a core component of the methodology, is not shown, and the figure is overly simplified.
2. Does the extreme accuracy distribution shown in Figure 1 actually occur? This may mislead readers into thinking that filtering out low-accuracy heads would result in significant improvements. Furthermore, the caption's expression "sum of their accuracies" is ambiguous and inconsistent with Algorithm 1. This needs to be expressed more precisely.
3. The caption of Figure 3 is cut off due to misuse of vspace.
4. What metric can be used to detect ensemble collapse?

**Audience:**

Yes

**Audience Explanation:**

Inference-time adaptation and reward model personalization are topics of high interest in the current LLM community, and a practical approach is relevant to both researchers and practitioners considering real-world deployment. In particular, the results demonstrating strong performance on RewardBench and efficiency compared to existing fine-tuning and few-shot prompting approaches are sufficient to attract the attention of TMLR readers interested in efficient ensemble-based adaptation methods, despite the lack of theoretical rigor.

**Claims And Evidence:**

No

**Claims Explanation:**

1. In Section 3.3, the generalized Bayesian inference interpretation consists of only two short paragraphs and provides only a superficial understanding. In particular, while mentioning "mild conditions," the authors do not verify what specific conditions are satisfied. They claim theoretical guarantees under the i.i.d. assumption, but there is insufficient discussion of the discrepancy between the theoretical guarantees and the actual implementation. Furthermore, it is unclear whether the reference [30] cited by the authors actually supports their claims.

2. Sec.4 is titled "When is Ensemble Reweighting Effective, and Why?" However, while it demonstrates why the method is effective, it does not clearly answer when it is effective. Moreover, there is no verification of whether the output distributions of ensemble heads actually approximate the Gumbel distribution.

3. HyRe utilizes N labeled examples from the target distribution, while the baseline uses zero, making it impossible to distinguish whether the performance improvement is due to the superiority of the method or simply due to the utilization of additional information.

4. Appendix E discusses the problems of repulsion-based diversity methods and claims that random priors are better. However, there are no comparative experimental results supporting this claim anywhere in the paper.

5. Appendices D and E do not substantially contribute to supporting the core contributions of the paper. Rather, they seem to reveal that important ablations that should have been conducted were not performed

**Requested Changes:**

1. The first sentence of the abstract needs to be reviewed to determine whether it is relevant to what this paper intends to convey.
2. Section 3.3 needs to be strengthened to address the concerns I raised in the supported claims section.
3. In Section 3.3, the statement "This makes HyRe particularly suitable for robust adaptation with non-differentiable metrics" needs to specify what exactly HyRe is robust to. If the authors can provide quantitative definitions or measurements related to robustness, this would better support their claims.
4. Regarding the diversity coefficient in Figure 5, it is difficult to determine which coefficient was used in the actual experiments in Section 6.2.
5. The variable $p$ in Algorithm 1 is not defined at all. Although it refers to Section 3.2, there is no explanation of $p$ there.
6. The notation inconsistency between Line 5 of Algorithm 1 and Equation (1) is confusing. This needs to be corrected or clarified.
7. In Algorithm 1, $c(x_n)$ is not defined. If the step is optional, it is unclear where $y_n$​ comes from when this step is skipped. Additionally, the two modes, depending on the option, are not clearly distinguished or explained. Furthermore, it is unclear whether the notation for the number of heads should be $H$ or $K$.
8. In the Key Insight box of Section 4, the expression "right combination" is ambiguous and needs improvement.
Can the authors provide a sensitivity study on the effect of ensemble size $K$?
9. Can the authors provide a failure case analysis addressing specific examples and causes where HyRe fails? The WILDS failure results are provided only in the Appendix; they should be brought into the main text and discussed further.
10. There are missing references to the Appendix, and the organization and structure of the paper need to be reviewed. Important information that should be in the main text exists only in the Appendix, or is missing from the Appendix entirely.
11. Although Appendix E discusses specific methods, readers may find it difficult to understand why methods that are not used (Vanilla, Epinet, Repulsion) are explained at length in the Appendix. It would be better to place this discussion in the main text and refer to the Appendix for details.

---

> ### Author Response · Authors · 2026-02-16
> **Response to Reviewer z4ps**
>
> We thank the reviewer for the detailed feedback.
>
> > "HyRe [uses] N labeled examples... making it impossible to distinguish whether the performance improvement is due to the superiority of the method or simply due to the utilization of additional information."
>
> The paper includes controlled comparisons at the same N: few-shot prompting (Table 4; HyRe outperforms at every N), fine-tuning on the HH target splits (Table 2), and 4 ensemble weighting methods (Table 8). Table 3 compares against RewardBench leaderboard scores (N=0 by convention) because no established methods exist for few-shot reward model adaptation.
>
> > "While mentioning 'mild conditions,' the authors do not verify what specific conditions are satisfied."
>
> We have rewritten Section 3.3. Bissiri et al. (2016) establish that the generalized Bayesian update is the unique coherent belief-updating procedure under basic rationality axioms. Concentration on expected-loss minimizers holds under i.i.d. data and bounded loss, both satisfied in our setup. When adaptation data comes from a shifted target, the i.i.d. assumption still holds, but guarantees apply only relative to the best head.
>
> > "Sec.4... does not clearly answer when [ensemble reweighting] is effective."
>
> We added a summary paragraph to Section 4 making the conditions explicit: HyRe is effective when (1) ensemble diversity captures task ambiguity, (2) conflicting preferences exist that individual heads can resolve, and (3) adaptation data is scarce. It is less effective under severe covariate shift (WILDS, Section 7) or abundant data (where fine-tuning wins).
>
> > "Ensemble diversity is assumed rather than guaranteed..."
>
> Diversity is maintained by prior networks (Osband et al., 2018), which are fixed and randomly initialized and added to each head's output. No existing deep ensemble method provides formal diversity guarantees. Our empirical evidence is strong: (a) prior scale controls per-head sharpness (Figure 4), (b) gains across 32 datasets (Figure 5) would be impossible under collapse, and (c) the WILDS failure case confirms diversity has real limits.
>
> > "There is no verification of whether the output distributions of ensemble heads actually approximate the Gumbel distribution."
>
> Section 4 does not claim this, nor is it necessary for HyRe to work. The Gumbel connection is a standard property of the Bradley-Terry model: any BT distribution can be decomposed into deterministic decision-makers with Gumbel utilities. This motivates why a diverse ensemble can capture a population of annotators.
>
> > "The WILDS failure results... should be brought into the main text."
>
> Thank you for the suggestion. Section 7 now explicitly reports that, across four WILDS datasets, reweighting did not outperform the uniform ensemble. We hypothesize that this is due to these shifts being structural rather than preference-based, and believe future work can investigate the specific dependence on the nature of the distribution shift.
>
> > "Can the authors provide a sensitivity study on the effect of ensemble size K?"
>
> We subsample $K \in \{5, 10, 25, 50, 100\}$ heads and evaluate on RewardBench without retraining. Most improvement comes from $K{=}5$ to $K{=}25$, with diminishing returns beyond $K \approx 50$.
>
> | $K$ | $N{=}0$ | $N{=}1$ | $N{=}5$ | $N{=}10$ | $N{=}20$ |
> |-----|---------|---------|---------|----------|----------|
> | 5   | 89.94±1.88 | 90.50±1.56 | 90.93±1.36 | 90.95±1.36 | 90.89±1.39 |
> | 10  | 91.13±1.06 | 91.62±0.82 | 92.10±0.68 | 92.23±0.68 | 92.08±0.73 |
> | 25  | 92.20±0.55 | 92.47±0.49 | 92.79±0.42 | 93.02±0.43 | 93.03±0.43 |
> | 50  | 92.33±0.40 | 92.60±0.33 | 92.96±0.28 | 93.26±0.28 | 93.41±0.29 |
> | 100 | 92.40±0.14 | 92.70±0.13 | 93.08±0.14 | 93.44±0.13 | 93.67±0.14 |
>
> We also report $K_\text{eff}$ (perplexity of the weight distribution). On average 43 of 100 heads carry meaningful weight; datasets with lower $K_\text{eff}$ show the largest gains from reweighting.
>
> > "The authors present 'sub-second adaptation for millions of concurrent users' as the motivation..."
>
> Section 6.4 directly supports this claim. Reweighting requires one forward pass with <1% overhead, and per-user state is a weight vector of $K$ numbers (0.03% parameter overhead).
>
> > "The notation inconsistency between Line 5 of Algorithm 1 and Equation (1)..."
>
> Thank you for pointing this out. Algorithm 1 now uses consistent notation: loss matches Equation (1), $Q$ is explicitly updated, and indices use $M$ to avoid collision with $N$.

---

> ### Comment · Reviewer_z4ps · 2026-02-18
>
> I thank the authors for their responses. Most of my concerns have been addressed. However, two minor points remain:
>
> - The Takeaway in Section 3.3 does not distinguish which types of distribution shift HyRe is robust to versus which it is not.
> - Section 3.3 emphasizes the i.i.d. assumption, but the authors should briefly acknowledge that their method uses active sampling in practice and discuss why this does not affect performance.

---

> > ### Author Response · Authors · 2026-02-18
> > **Response to Reviewer z4ps (Follow-up)**
> >
> > Thank you for following up with your remaining concerns. We have additionally addressed both points in the revised submission (newer changes in **blue**):
> >
> > 1. The Takeaway in Section 3.3 now distinguishes preference-based shift (effective) from structural covariate shift (negative results).
> > 2. We added a sentence noting that active data selection, an optional variant, technically violates the i.i.d. assumption but only improves sample efficiency in practice.

---

> > > ### Comment · Reviewer_z4ps · 2026-02-19
> > >
> > > I've read the revised version with blue-highlighted sentences. My remaining concerns have been addressed by the authors' revisions and response. Thank you for actively accepting my feedback.

---

### Author Response · Authors · 2026-02-16
**General Response to All Reviewers**

We thank all reviewers for their detailed and constructive feedback. We summarize our main changes below, and respond to specific comments in individual replies.

**Source of gains.** We added a consolidated analysis (Section 6.4) showing that HyRe's gains are specific to Bayesian reweighting and outperform the uniform ensemble, few-shot prompting, fine-tuning, and four alternative weighting schemes at the same N.

**Narrowed scope and failure modes.** We narrowed the Introduction and Abstract to focus on personalization rather than general distribution shift, and added concrete WILDS failure cases, a K-ablation, and effective ensemble size measurements to Section 7, confirming that HyRe's applicability is scoped to settings where the ensemble spans the target behavior.

**Theoretical justification.** We rewrote Section 3.3 to clarify that the generalized Bayesian update is axiomatically justified as the unique coherent belief update given a loss function (Bissiri et al., 2016), with no identifiability or likelihood assumptions required, and stated when concentration guarantees apply.

**Additional changes:** added temperature sensitivity study; fixed Algorithm 1 notation; added Ethics/Reproducibility statements; added RISE-Judge baseline; restructured appendices.

---

### Decision · Action_Editor_cAfP · 2026-03-26

**Recommendation:** Accept as is

**Audience:**

Yes

**Audience Explanation:**

This paper provides a practical and timely contribution to inference-time adaptation for reward models, which may be of interest to a subset of the TMLR audience.

**Claims And Evidence:**

Yes

**Claims Explanation:**

The paper provides a clear empirical contribution and, following revision, the claims are appropriately scoped to settings in which the proposed method performs well, namely inference-time personalization. The rebuttal and revision address the main concerns raised by the reviewers by (i) clarifying the scope of applicability, including negative results under structural covariate shift, (ii) strengthening the discussion of the generalized Bayesian interpretation, and (iii) adding ablations demonstrating that the observed gains are specific to Bayesian reweighting. The resulting evidence is convincing for the revised claims, even if the novelty is more practical and incremental than fundamentally conceptual.